# TADA: Improved Diffusion Sampling with Training-free Augmented DynAmics

**Tianrong Chen, Huangjie Zheng, David Berthelot, Jiatao Gu, Josh Susskind, Shuangfei Zhai**
Apple
{tchen54,huangjie_zheng,dberthelot,jgu32, jsusskind,szhai}@apple.com

## Abstract

Diffusion models have demonstrated exceptional capabilities in generating high-fidelity images but typically suffer from inefficient sampling. Many solver designs and noise scheduling strategies have been proposed to dramatically improve sampling speeds. In this paper, we introduce a new sampling method that is up to 186% faster than the current state of the art solver for comparative FID on ImageNet512. This new sampling method is training-free and uses an ordinary differential equation (ODE) solver. The key to our method resides in using higher-dimensional initial noise, allowing to produce more detailed samples with less function evaluations from existing pretrained diffusion models. In addition, by design our solver allows to control the level of detail through a simple hyper-parameter at no extra computational cost. We present how our approach leverages momentum dynamics by establishing a fundamental equivalence between momentum diffusion models and conventional diffusion models with respect to their training paradigms. Moreover, we observe the use of higher-dimensional noise naturally exhibits characteristics similar to stochastic differential equations (SDEs). Finally, we demonstrate strong performances on a set of representative pretrained diffusion models, including EDM, EDM2, and Stable-Diffusion 3, which cover models in both pixel and latent spaces, as well as class and text conditional settings. The code is available at https://github.com/apple/ml-tada.

## 1 Introduction

Diffusion Models (DMs; Song et al. [27]; Ho et al. [14]) and Flow Matching [20, 21] are foundational techniques in generative modeling, widely recognized for their impressive scalability and capability to generate high-resolution, high-fidelity images [4, 10, 24]. Both share a common underlying mathematical structure and generate data by iteratively denoising Gaussian noise through a solver.

Sampling from a diffusion model is typically discretized through a multi-step noise schedule. Denoising Diffusion Probabilistic Models sample new noise at every step resulting in a process interpretable as discretized solutions of diffusion Stochastic Differential Equations (SDEs). Meanwhile, Denoising Diffusion Implicit Models only sample initial noise which is reused at every step resulting in a process interpretable as discretized solutions of Ordinary Differential Equations (ODEs). Both ODE and SDE solvers should perform similarly since they merely represent different interpretations of the same Fokker-Planck Partial Differential Equation. In practice, however, ODE solvers often yield lower-fidelity results than SDE solvers as evidenced by the FID scores in [17] when sufficient function evaluations are performed and model capacity is a limiting factor. On the other hand, SDE solvers require finer steps due to the noise injection at every step, which gets amplified through discretization.

Generating high-quality images using DMs often necessitates a high number of steps and therefore a high Number of Function Evaluations (NFEs), substantially increasing computational costs com-

39th Conference on Neural Information Processing Systems (NeurIPS 2025).

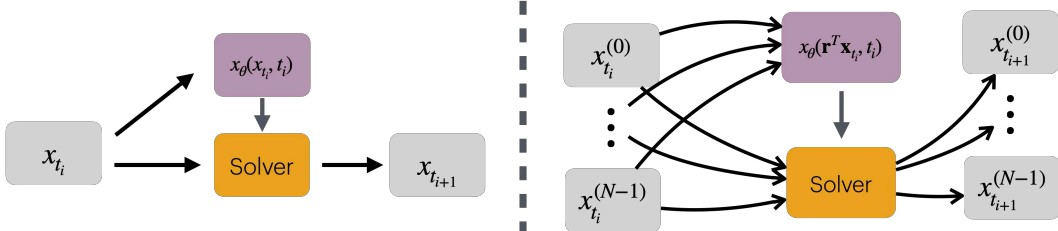

Figure 1: Here we show the distinctions between conventional diffusion models (*left*) and momentum diffusion models (*right*) during sampling. Leveraging Prop 3.1, we demonstrate that pretrained diffusion models with $x_0$-*prediction* $x_\theta(\cdot, \cdot)$ can be directly applied to propagate the momentum system with multiple varible $\{\mathbf{x}_{t_i}^{(}n)\}_0^{N-1}$. Moreover, the choice of numerical solver remains flexible. See Sec 3.2 for details.

pared to alternative generative approaches such as generative adversarial networks (GANs) [11]. Consequently, a large research effort has been focused on reducing the NFEs to reach a desired level of sample quality and this is also the focus of our paper. Recent efforts [17, 32] have significantly improved the efficiency of SDE solvers, achieving compelling results with a moderate number of function evaluations. Designing training-free faster solvers by leveraging numerical integration techniques has also received a lot of attention: exponential integrators [15] and multi-step methods [1], and hybrid combinations of these [22, 23, 33, 34]. Remarkably, faster solvers reach near-optimal quality with around 30 NFEs and retain acceptable image generation capabilities even with NFE $< 10$.

To further enhance performance, optimization-based solvers [35, 36] have advanced the capabilities of the aforementioned methods, achieving strong results with as few as 5 NFEs. In parallel, training-based dynamical distillation approaches [3, 25, 28, 30] have demonstrated effectiveness in reducing the number of function evaluations and can similarly benefit from the incorporation of fast solvers.

In this paper, we propose a new sampling method that is 186% faster than the current state of the art for generating 512x512 ImageNet samples with an FID of 2. Our method is orthogonal to existing diffusion model sampling techniques, allowing seamless integration with advanced solvers and classifier-free guidance (CFG) schedules simply by transitioning from standard ODE to momentum ODE frameworks. The proposed method is based on momentum diffusion models [6] and on the observation [7] that they can achieve competitive performance with a very low number of function evaluations. Specifically our paper extends the aforementioned works as follows:

1. We prove the training equivalence between momentum diffusion models [6, 9] and conventional diffusion models, modulo a transformation of the input variables to the neural network. Consequently, the equivalence shows that simple noise or noisy data augmentation together with diffusion loss, including but not limited to momentum diffusion, offers **no training benefit**. Readers may extend this reasoning to their own settings and verify the algorithms relevant to them.

2. We then shift our attention to the **sampling phase** of the momentum system. We propose a new sampling method named Training-free Augmented DynAmics (TADA) that uses higher-dimensional input noise and yet enables direct reuse of pretrained diffusion models.

3. Although no benefits are observed in the training phase in theory, we find that our proposed method, rooted in the momentum system, inherits advantageous SDE properties while employing ODE solvers, enabling both diverse generation and accelerated sampling through momentum dynamics.

4. We illustrate the superior performance of our approach through extensive experimentation.

## 2    Preliminary

First, we cover conventional diffusion models and flow matching, we then follow up by introducing momentum diffusion models and we finish this section with exponential integrators. In the rest of this paper, we follow the flow matching literature convention for the direction of distribution transport, e.g. from the prior distribution at time $t = 0$ to the data distribution at time $t = 1$.

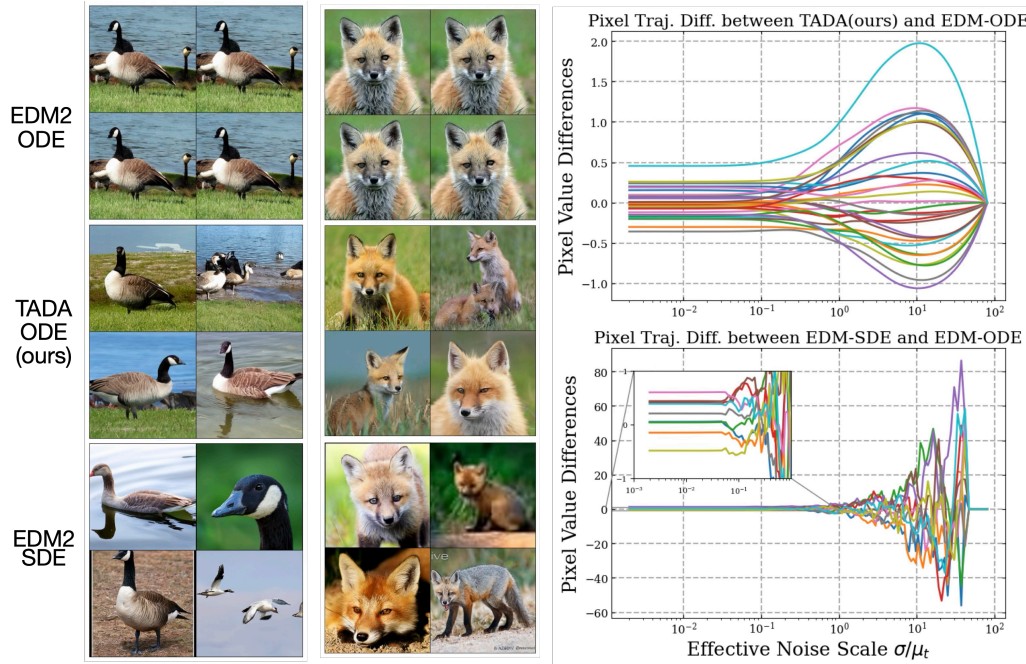

Figure 2: Demonstration of differences of TADA and probablistic ODE in terms of trajectories. We apply the same pretrained model with momentum system with same discretization in terms of SNR and the same initial prior samples. TADA can generate SDE-like property, such as generate different samples from same initial condition, but the system keeps deterministic ODE which can be solved more efficiently. See Prop.3.3 for more detail.

## 2.1 Diffusion Models and Flow Matching

In Diffusion Models (DM) and Flow Matching (FM), intermediate states $x_t \in \mathbb{R}^d$ are sampled from a tractable transition probability conditioned on an initial data point $x_1 \sim p_{\text{data}}$ during training:

$$x_t = \mu_t x_1 + \sigma_t \epsilon, \quad \epsilon \sim \mathcal{N}(0, \boldsymbol{I}_d). \tag{1}$$

The coefficients $\mu_t \in \mathbb{R}$ and $\sigma_t \in \mathbb{R}$ differ depending on the specific model parameterization: mean preserving, variance preserving or variance exploding. Regardless of these differences, the training objectives across various methods remain identical.

For the sake of clarity, we illustrate parameterizing the *noise prediction model* with a neural network with learnable weights $\theta$. In this case the objective function can be expressed as:

$$\min_{\theta} \mathcal{L}_{\text{DM}}(\theta) := \mathbb{E}_{x_1, \epsilon, t} \|\epsilon_\theta(x_t, t) - \epsilon\|_2^2.$$

Alternatively, using the linear relationship given by eq.1, we could just as easily parameterize the *data prediction model* as $x_\theta(x_t, t) := (x_t - \sigma_t \epsilon_\theta(x_t, t))/\mu_t$, or even as a velocity prediction model.

Without loss of generality, we define a force term $F_\theta : \mathbb{R}^d \times [0, 1] \to \mathbb{R}^d$ as a linear combination of the learned noise $\epsilon_\theta$ and the state $x_t$, which pushes forward $x_t$ from the prior towards the $p_{\text{data}}$ from $t = 0$ to $t = 1$. During sampling, we use this force $F_\theta$ to define the following probabilistic flow:

$$\frac{\mathrm{d}x_t}{\mathrm{d}t} = a_t x_t + b_t F_\theta(x_t, t), \quad x_0 \sim \mathcal{N}(0, \sigma_0^2 \boldsymbol{I}_d), \tag{2}$$

where $a_t \in \mathbb{R}$, and $b_t \in \mathbb{R}$ are time-varying coefficients defined by specific parameterization of $F_\theta$. For example, for a FM parameterization: $F_\theta(x_t, t) = (x_\theta(x_t, t) - x_t)/(1 - t)$, $a_t = 0$ and $b_t = 1$.

## 2.2 Phase Space Momentum Diffusion Model

Several recent studies have explored a more sophisticated diffusion process defined in the phase space [7, 9]. In this space there are $N = 2$ noise variables, consequently $\mathbf{x}_t, \boldsymbol{\epsilon} \in (\mathbb{R}^d)^2$. Owing to the linear

dynamics within this framework—similar to those in conventional diffusion models—the transition probability can also be derived analytically:

$$\mathbf{x}_t = (\boldsymbol{\mu}_t \otimes \mathbf{I}_d)x_1 + (\boldsymbol{L}_t \otimes \mathbf{I}_d)\boldsymbol{\epsilon}, \quad \boldsymbol{L}_t\boldsymbol{L}_t^\mathsf{T} = \boldsymbol{\Sigma}_t, \quad \boldsymbol{\epsilon} := \left[\epsilon^{(0)}, \epsilon^{(1)}\right]^\mathsf{T} \sim \mathcal{N}(0, \boldsymbol{I}_{2d}),$$

where $\boldsymbol{\mu}_t \in \mathbb{R}^2$ and $\boldsymbol{\Sigma}_t \in \mathbb{R}^{2\times2}$ denote the mean and covariance matrix of the resulting multi-variable Gaussian distribution. $\boldsymbol{L}_t \in \mathbb{R}^{2\times2}$ is the noise scaling matrix. The superscript denotes the $i$-th variable in the multi-variable setting, while the boldface notation $\mathbf{x}_t$ represents the aggregation of variables, i.e., $\mathbf{x}_t = [x_t^{(0)}, x_t^{(1)}]^\mathsf{T}$ in this setting.

**Notation**: In the rest of this document, we use of the Kronecker product to specify that all the weights are identical along the data dimension $d$. For example, the expression $(\mu_t \otimes \mathbf{I}_d)x_1$ denotes the stacking of the scaled versions of $x_1$ by $\boldsymbol{\mu}_t^{(n)}$, e.g. $(\boldsymbol{\mu}_t \otimes \mathbf{I}_d)x_1 = [\boldsymbol{\mu}_t^{(0)}x_1, \boldsymbol{\mu}_t^{(1)}x_1]^\mathsf{T}$.

The standard training objective for MDM minimizes the approximation error of $\epsilon^{(1)}$:

$$\min_\theta \mathcal{L}_{\text{MDM}}(\theta) := \mathbb{E}_{x_1,\boldsymbol{\epsilon},t}\|\epsilon_\theta(\mathbf{x}_t, t) - \epsilon^{(1)}\|_2^2 \quad \text{with } \epsilon_\theta : (\mathbb{R}^d)^2 \times [0,1] \to \mathbb{R}^d \qquad (3)$$

**Note**: It should be observed that such techniques require training since the function $\epsilon_\theta$ takes two data inputs and cannot reuse pre-trained conventional diffusion models where $\epsilon_\theta$ takes only one data input.

Just like for conventional diffusion models, various parameterizations of the learned noise $\epsilon_\theta$ remain viable. In particular, AGM[6] and CLD[9] adopt distinct formulations of $F_\theta$, defined as linear combinations of $\epsilon_\theta$ and $\mathbf{x}_t$, specifically constructed to guide $x_t^{(0)}$ from the prior distribution to the target data distribution. Consequently, sampling reduces to solving a similar probabilistic flow:

$$\frac{\mathrm{d}\mathbf{x}_t}{\mathrm{d}t} = (\boldsymbol{A}_t \otimes \mathbf{I}_d)\mathbf{x}_t + (\mathbf{b}_t \otimes \mathbf{I}_d)F_\theta(\mathbf{x}_t, t), \quad \mathbf{x}_0 \sim \mathcal{N}(0, \boldsymbol{\Sigma}_0). \qquad (4)$$

where $\boldsymbol{A}_t \in \mathbb{R}^{2\times2}$ and $\mathbf{b}_t \in \mathbb{R}^2$ denote time-dependent coefficients. Specifically, in the MDM framework, the function $F_\theta : (\mathbb{R}^d)^2 \times [0,1] \to \mathbb{R}^d$ accepts an aggregated input of two variables of dimension $d$ and outputs a single vector of dimension $d$. The resulting $d$-dimensional output is subsequently broadcasted across the $N = 2$ variables through the coefficient $\mathbf{b}_t$, a key distinction from conventional DM. The explicit formulations for $\boldsymbol{A}_t$ and $\mathbf{b}_t$ are detailed in Appendix.F.1.

## 2.3 Sampling with Exponential Integrators

Once a pretrained diffusion model $\epsilon_\theta$ is obtained, the dynamics specified by eq. 2 or 4 can be readily solved. A variety of advanced training-free fast sampling techniques exist, and many of them rely on exponential integrators [15] which presents as the form of eq.5:

$$x_t = \Phi(t,s)x_s + \underbrace{\int_s^t \Phi(t,\tau)b_\tau F_\theta(x_\tau, \tau)\mathrm{d}\tau}_{\text{Approximator } \Psi(s,t,x_s,F_\theta)}. \qquad (5)$$

Where $\Phi(\cdot,\cdot)$ is the transition kernel induced by $A_t$. Since the first linear component of the ODE can be analytically integrated via the transition kernel, the primary challenge lies in accurately approximating the second nonlinear integration, such as the neural network parameterized function $F_\theta$ over discretized time steps $s < t$.

To improve the accuracy of approximating the nonlinear integral in the second term, advanced ODE solvers are employed as approximators $\Psi(\cdot,\cdot,\cdot,\cdot)$ in eq.5. Higher-order single-step methods such as Heun's method [12] and multi-step explicit methods like Adams–Bashforth [5] have been widely adopted in prior works [17, 22, 23, 33], often in conjunction with exponential integrators. To further enhance accuracy, implicit schemes such as the Adams–Moulton method [2] have also been introduced [34]. The current state-of-the-art solvers integrate these various techniques, and the specific combinations along with the resulting algorithms are summarized in Appendix.F.3.

This work, however, is specifically focused on designing the ODE itself. Therefore, our approach is entirely orthogonal to existing solver methodologies, allowing it to be seamlessly integrated with any solver type, see fig.1 for explanation and demonstration.

**Algorithm 1** TADA sampling

---

**Require:** discretized times $t_i \in [t_0, t_1, ... t_T]$; ODE Solver as approximator $\Psi(\cdot, \cdot, \cdot, \cdot)$ (see eq. 5);
  Pretrained DM $x_\theta(\cdot, \cdot)$. Transition function: $\Phi(t, s) = \exp \int_s^t \mathbf{A}_\tau d\tau$. Cache $Q$.

1:  $\mathbf{x}_{t_0} \sim \mathcal{N}(0, \boldsymbol{\Sigma}_{t_0})$ $\qquad\qquad\qquad\qquad\qquad\qquad$ ▷ draw prior sample
2:  **for** $i = 0$ to $T - 1$ **do**
3:  $\qquad$ compute $\boldsymbol{\mu}_{t_i}, \boldsymbol{\Sigma}_{t_i}$, and $\mathbf{r}_{t_i} := \frac{\boldsymbol{\Sigma}_{t_i}^{-1} \boldsymbol{\mu}_{t_i}}{\boldsymbol{\mu}_{t_i}^{\mathsf{T}} \boldsymbol{\Sigma}_{t_i}^{-1} \boldsymbol{\mu}_{t_i}}$ $\qquad$ ▷ obtain the reweighting for $N$ variables
4:  $\qquad$ $\hat{x} \leftarrow x_\theta\big((\mathbf{r}_{t_i}^{\mathsf{T}} \otimes I_d) \mathbf{x}_{t_i}, t_i\big)$ $\qquad\qquad$ ▷ data prediction with pretrained DM
5:  $\qquad$ $F_\theta \leftarrow N! \dfrac{\hat{x} - \sum_{n=0}^{N-1} \frac{x_{t_i}^{(n)}}{n!}(1 - t_i)^n}{(1 - t_i)^N}$ $\qquad$ ▷ Compute force term, see Sec. 3.2
6:  $\qquad$ $\Psi_{t_i} \approx \displaystyle\int_{t_i}^{t_{i+1}} \Phi(t_{i+1}, \tau)\, \mathbf{b}_\tau F(\mathbf{x}_\tau, \tau)\, d\tau$ $\qquad$ ▷ Approx. nonliear part using existing solver (with $Q$ if solver is multistep)
7:  $\qquad$ **if** Solver is multistep **then** $Q \xleftarrow{\text{cache}} \hat{x}$ $\qquad$ ▷ store history
8:  $\qquad$ $\mathbf{x}_{t_{i+1}} \leftarrow \Phi(t_{i+1}, t_i) \mathbf{x}_{t_i} + \Psi_{t_i}$ $\qquad\qquad$ ▷ state update
9:  **end for**
10: **return** $x_\theta\big((\mathbf{r}_{t_T}^{\mathsf{T}} \otimes I_d)\, \mathbf{x}_{t_T},\, t_T\big)$ $\qquad\qquad$ ▷ return results with data prediction

---

## 3  Method

We start in Sec. 3.1 with a proof for the training equivalence between momentum diffusion models and conventional diffusion models for any $N \geq 1$. This condition is necessary both for the training-free property of our method as well as for the generalization to arbitrarily large $N$. We then introduce the proposed method itself in Sec. 3.2. Finally in Sec. 3.3, we study the dynamics of the proposed method and how they tie SDE and ODE formulations.

### 3.1  Training equivalence between Momentum Diffusion Models and Diffusion Models

In Momentum Diffusion Model case, typically, neural networks are parameterized with $N \in \{2, 3\}$ augmented variables as input. This method naively introduce certain problems. While state-of-the-art diffusion models have developed advanced signal-to-noise ratio (SNR) schedules achieving excellent results [16, 17, 19], defining an appropriate SNR within momentum systems is challenging. Each variable in a momentum system inherently has its own SNR and is coupled via a covariance matrix, complicating the definition and practical usage of SNR. Consequently, this complexity has hindered progress in momentum-based diffusion modeling. Here, we show that the SNR of the momentum diffusion model can be characterized as the optimal SNR achievable through a linear combination of multiple variables, as stated in the following proposition.

**Proposition 3.1.** *The training objective of general Momentum Diffusion Models (MDM) (i.e., eq. 3) can be equivalently reparameterized as:*

$$\mathcal{L}_{MDM}(\theta) \propto \mathbb{E}_{\mathbf{x}_t} ||x_\theta(\mathbf{x}_t, t) - x_1||_2^2 \implies x_\theta^*(\mathbf{x}_t, t) = \mathbb{E}[x_1 | \mathbf{x}_t]$$

$$\mathbb{E}[x_1 | \mathbf{x}_t] = \mathbb{E}\big[x_1 | (\mathbf{r}_t^{\mathsf{T}} \otimes \mathbf{I}_d)\mathbf{x}_t, \boldsymbol{\epsilon}\big] = \mathbb{E}\big[x_1 | (\mathbf{r}_t^{\mathsf{T}} \otimes \mathbf{I}_d)\mathbf{x}_t\big] \quad \text{where } \mathbf{r}_t := \frac{\boldsymbol{\Sigma}_t^{-1} \boldsymbol{\mu}_t}{\boldsymbol{\mu}_t^{\mathsf{T}} \boldsymbol{\Sigma}_t^{-1} \boldsymbol{\mu}_t}.$$

Moreover, the $F_\theta$ in eq. 4 can be recovered as a linear combination of $x_\theta$ and $\mathbf{x}_t$ (see section.3.2 for details). Here, $\boldsymbol{\mu}_t^{\mathsf{T}} \boldsymbol{\Sigma}_t^{-1} \boldsymbol{\mu}_t$ is the effective SNR of $(\mathbf{r}_t^{\mathsf{T}} \otimes \mathbf{I}_d)\mathbf{x}_t$ which simply is a weighted linear combination of $\mathbf{x}_t$ by $\mathbf{r}_t$.

*Proof.* See Appendix. B. $\qquad\qquad\qquad\qquad\qquad\qquad\qquad\qquad\qquad\qquad\qquad\qquad\qquad$ □

This proposition holds significant implications despite its apparent conceptual simplicity: It demonstrates that MDM, even when involving multiple input variables to the neural network, can be trained using a single input constructed as a $\mathbf{r}_t$-weighted linear combination of the $N$ input variables. As a consequence, the training objective becomes equivalent to that of a conventional diffusion model, addressing debates about potential advantages in training arising from momentum diffusion or trivial

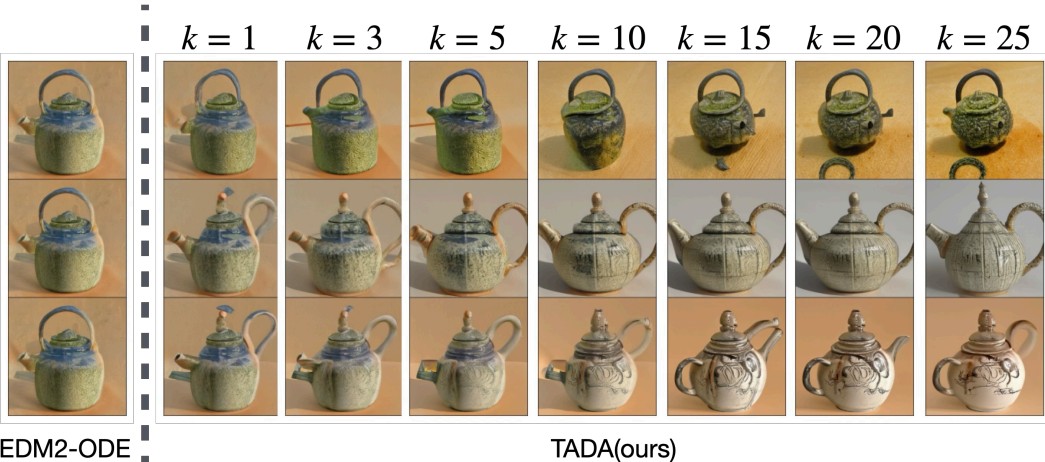

EDM2-ODE       TADA(ours)

Figure 3: Generated samples under varying prior scales are shown, all initialized with the same initial condition of dynamics in Prop. 3.3:$y_0 := (\mathbf{r}_0^\top \otimes \boldsymbol{I}_d)\mathbf{x}_0 \equiv \epsilon$, using an identical time discretization over the SNR and the same pretrained model with 15 NFEs. It can be observed that the diversity of the generated results increases proportionally with the standard deviation of the final variable $x_0^{(N-1)}$, which is scaled by a factor $k$: $\boldsymbol{\Sigma}_0 = \mathrm{diag}(1, 1, ..., k)$.

variable augmentations. Importantly, this also allows direct reuse of pretrained conventional diffusion models within the MDM framework and therefore the training-free property claimed by our method.

## 3.2 Momentum Diffusion Sampling Methodology

[6] highlighted that MDM can yield promising results with small numbers of function evaluations (NFE); however, the absolute performance at sufficient NFE lags behind traditional diffusion models, revealing fundamental training issues. Due to the observation in Prop. 3.1, such training issues can be trivially resolved, or, even more easily, we can simply plug in the pretrained diffusion model.

**Training-free Augmented DynAmics (TADA)**

We now present the sampling procedure after plugging the pretrained diffusion model into the system. Now the input of neural network becomes $(\mathbf{r}_t^\top \otimes \mathbf{I}_d)\mathbf{x}_t \in \mathbb{R}^d$ instead of $\mathbf{x}_t \in (\mathbb{R}^d)^N$ as is the case in vanilla MDM with $N = 2$. Since $\boldsymbol{\mu}_t^\top \boldsymbol{\Sigma}_t^{-1} \boldsymbol{\mu}_t$ is the effective SNR in the momentum system, one can simply map it to the time conditioning used in the pretrained model.

We extend the AGM [6] framework to an arbitrary $N$-variables augmented space. Similarly to Sec. 2.2, we reparameterize the data estimation $x_\theta$ to the force term $F_\theta$, which drives the dynamics of $x_t^{(0)}$ toward $x_1^{(0)} \sim p_{\text{data}}$ explicitly. The matrices involved have the closed form:

$$\mathbf{A}_t = \begin{bmatrix} 0 & 1 & 0 & \cdots & 0 \\ 0 & 0 & 1 & \cdots & 0 \\ \vdots & \vdots & \ddots & \ddots & \vdots \\ 0 & 0 & \cdots & 0 & 1 \\ 0 & 0 & \cdots & 0 & 0 \end{bmatrix}_{N\times N}, \mathbf{b}_t := \begin{bmatrix} 0 \\ 0 \\ \vdots \\ \vdots \\ 1 \end{bmatrix}; F_\theta(\mathbf{x}_t, t) := N! \frac{x_\theta(\mathbf{r}^\top \mathbf{x}_t, t) - \sum_{n=0}^{N-1} \frac{x_t^{(n)}}{n!}(1-t)^n}{(1-t)^N}.$$

**Remark 3.2.** *When $N = 1$, this formulation simplifies precisely to vanilla flow matching. When $N = 2$, it is essentially same as the deterministic case of AGM [6] but with the added training-free property that our method carries. Please see Appendix. F.4 for details.*

To compute the reweighting term $\mathbf{r}_t$, the mean and covariance matrix at time $t$ are required. These quantities can be obtained by analytically solving the coupled dynamics of the mean $\boldsymbol{\mu}_t$ and covariance $\boldsymbol{\Sigma}_t$, a standard approach [26] in both conventional DM and MDM. Due to space constraints, the explicit analytical expressions are presented in Appendix F.2.

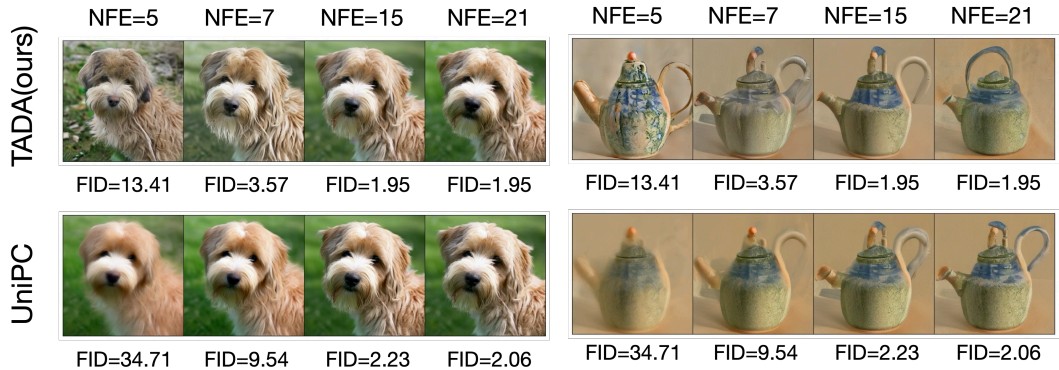

Figure 4: Qualitative comparison with UniPC, varying the NFEs, using the same initial condition and the same pretrained EDM2 model. More qualitative comparision can be found in Appendix. G.

## 3.3 Analysis of Sampling Dynamics

In conventional diffusion models, variations in generative dynamics largely stem from differences in time discretization schemes, as discussed in [17]; for example, Variance Preserving, Variance Exploding, and Flow Matching dynamics primarily differ in their time discretization over SNR during sampling. This naturally raises the question of whether momentum diffusion simply constitutes another form of time discretization. To address this, we conduct a detailed analysis of variable $y_t$ which is fed into the neural network in Prop. 3.3.

**Proposition 3.3.** *The dynamics of neural network input* $y_t := (\mathbf{r}_t^\top \otimes \boldsymbol{I}_d)\mathbf{x}_t$ *is given by:*

$$\frac{\mathrm{d}y_t}{\mathrm{d}t} = \underbrace{\sum_{i=0}^{N-1} w_t^{(i)} \epsilon^{(i)}(x_t^{(i)})}_{Pseudo\ Noise} + \alpha_t y_t + \beta_t x_\theta(y_t, t).$$

*For the coefficient of* $w_t^i$, $\alpha_t$ *and* $\beta_t$, *please refer to Appendix. C.* $\epsilon_t^{(i)}$ *denotes the estimated Gaussian noise for each variable* $x_t^i$, *induced by* $x_\theta$. *This is analogous to the standard diffusion model but extended to the multi-variable case. Please see Appendix.C for more detail.*

*Proof.* See Appendix. C. $\qquad\square$

**Remark 3.4.** *There are two scenarios in which Prop 3.3 degenerates into a mere different time discretization of the conventional diffusion model, irrespective of the value of* $N$. *The first occurs when* $N = 1$, *and the second arises when* $\mathbf{A}_t$ *is a diagonal matrix, implying that each variable evolves independently. Further details and proof are provided in Appendix. F.5.*

As demonstrated in Prop 3.3, the dynamics of the neural network input $y_t$ cannot be expressed solely as a function of $y_t$; rather, an additional *pseudo noise* term emerges due to interactions among variables, endowing the system with SDE properties, even when solving the deterministic ODE in Eq. 4. Notably, we empirically find that, the diversity of samples generated from the same prior can be explicitly manipulated by scaling the standard deviation of the final variable $x_0^{(N-1)}$ by a factor of $k$, corresponding to the last diagonal entry of the covariance matrix $\boldsymbol{\Sigma}_0$, as illustrated in Fig. 3.

## 4 Experiments

In this section, we evaluate the performance TADA in comparison with a range of ODE and SDE solvers based on conventional first-order diffusion probabilistic flows. Specifically, we assess diffusion models such as EDM [17], EDM2 [18], in both pixel and latent spaces on the ImageNet-64 and ImageNet-512 datasets [8]. Additionally, we evaluate the flow matching model Stable Diffusion 3 [10] in the latent space.

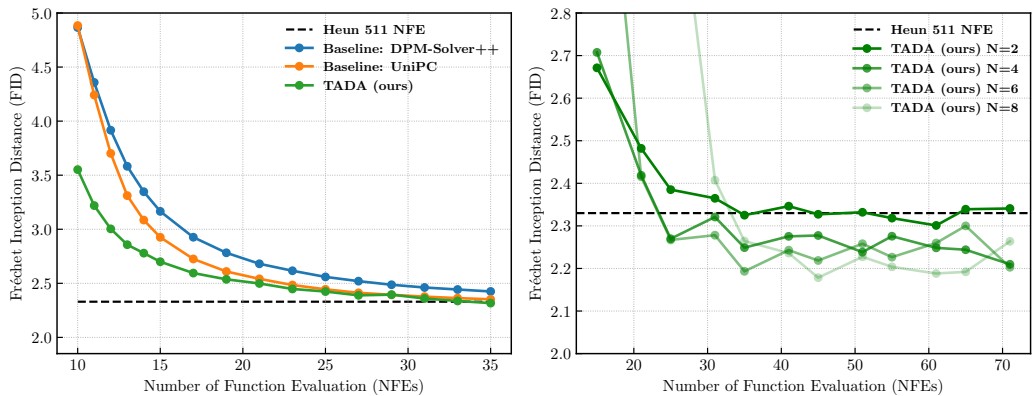

Figure 5: *left*: Comparison with baselines on ImageNet-64 using EDM pretrained model. *Right*: Performance under varying numbers of variables $N$ while keeping the SNR-based time discretization same to the $N = 2$ setting; $N = 2$ is the default configuration reported throughout this paper.

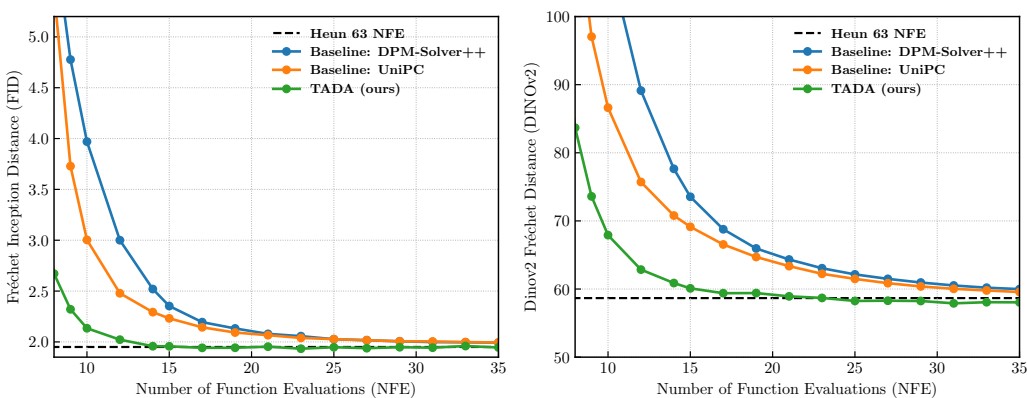

Figure 6: Comparison with baselines on ImageNet-512 with EDM2 pretrained model.

### 4.1   EDM and EDM2 Experiment

We begin by benchmarking our method against UniPC [34] and DPM-Solver++ [23] on both ImageNet-64 and ImageNet-512, using the EDM [17] and EDM2 [18] frameworks. To ensure fair and competitive evaluation, we conduct comprehensive ablations over all combinations of discretization schemes, solver variants, and solver orders available in the respective codebases, selecting the best-performing configurations for comparison. Full ablation results for the baselines are reported in the Supplementary Material. For our method, TADA , we consistently employ the simplest multi-step exponential integrator with third-order solvers, using the same polynomial discretization across all experiments.

We evaluate performance using Fréchet Inception Distance score(FID [13]) for both ImageNet-64 and ImageNet-512, and additionally use Fréchet Distance-DINOv2 (FD-DINOv2 [29]) for ImageNet-512. Our results show that TADA consistently outperforms the baselines across all tested numbers of function evaluations (NFEs) in fig. 5 and fig. 6. We also examine the case with $N > 2$, which introduces additional pseudo noise as described in Prop. 3.3. In all setups, we use the same discretization over SNR and vary only the standard deviation of $x_0^{(N-1)}$. This configuration results in improved performance on ImageNet-64, as shown in fig 5. However, no consistent improvement is observed on ImageNet-512, which may be attributed to the high capacity of the neural network and limited exploration of time discretization, thereby diminishing the impact of the additional noise perturbation. For consistency, we therefore report results only for the $N = 2$ in fig. 6 and the rest of experiments.

Fig 4 presents qualitative results under varying NFE budgets. Unlike conventional ODE solvers, which tend to produce increasingly blurred outputs as NFEs decrease, TADA continues to generate

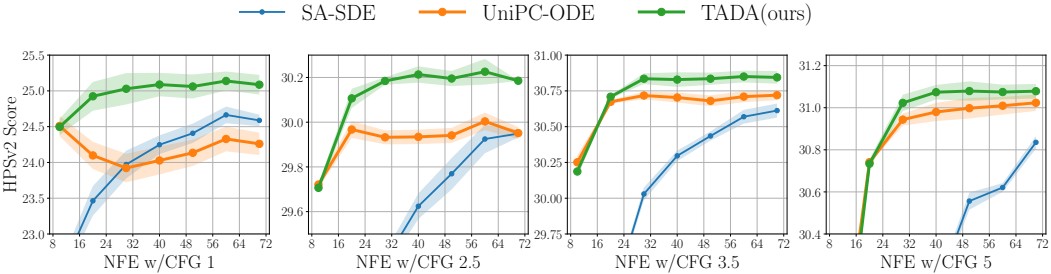

Figure 7: Comparison with baselines with HpsV2 metrics with SD3 pretrained model.

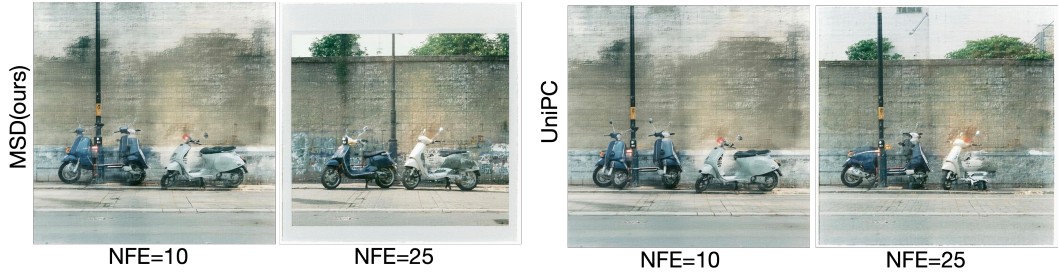

Figure 8: Qualitative Comparison with baselines with SD3 model w/o CFG. UniPC delivers no additional gains in image quality, whereas our method continues to improve. This trend is evident both in Fig. 7 and in the samples generated at NFE = 25.

plausible images with preserved details. Nevertheless, these images may still deviate from the data distribution $p_{\text{data}}$, as indicated by the larger FID metric. See Appendix. G for more examples.

## 4.2 SD3 Experiment

For evaluations on Stable Diffusion 3 [10], we exclude DPM-Solver++ from comparison, as UniPC has consistently outperformed it in Section 4.1. Additionally, we observe that the default Flow Matching solver in SD3 achieves performance comparable to UniPC. To enrich our baseline set, we include a recently proposed SDE-based solver, Stochastic Adam (SA) [32]. All baseline configurations follow the recommended settings from the official Diffusers implementation. For evaluation, we adopt HPSv2 [31] as our primary benchmark. Specifically, we generate images for all benchmark prompts, producing 3200 samples per seed, and report the mean and standard deviation over three random seeds, thus, totally 9600 images. To ensure consistency, our method uses the same SNR-based discretization scheme as SD3, which is also the default across all baseline implementations. For the sake of consistency, we employ a second-order multi-step solver, regardless of CFG scale.

Fig. 7 presents the quantitative performance of TADA compared to the baselines. Our proposed method consistently outperforms the baselines across all CFG strengths. However, the performance gap narrows as the CFG increases, suggesting that TADA is particularly effective at lower CFG values, where it achieves more pronounced improvements over existing methods. Fig 8 demonstrates that TADA produces images with more semantically coherent content at 25 NFEs than the baseline, an observation further supported by the quantitative metrics presented in Fig. 7.

## 5 Conclusions and Limitations

In this paper, we have identified and elucidated the underlying training principle of the momentum diffusion model, demonstrating that it fundamentally aligns with that of conventional diffusion models. This observation enables the direct integration of pretrained diffusion models into momentum-based systems without additional training. Furthermore, we conducted a thorough analysis of the sampling dynamics associated with this approach and discovered that the implicit system dynamics introduce additional pseudo-noise. Empirical evaluations confirmed that this characteristic indeed enhances the sampling quality across various datasets and pretrained models.

However, our approach also has limitations. On the theoretical side, Prop 3.3 does not disentangle the degrees of freedom associated with the pseudo noise dimension $N$ and the weighting coefficients $w_t^{(i)}$. As a result, while Fig.5 demonstrates that increasing system stochasticity can lead to improved outcomes, our method still falls short of matching the SDE results reported in [17] with 511 NFEs. This shortfall stems from the limited control over the injected stochasticity due to the naive choice of $\mathbf{A}_t$. On the practical side, experiments with SD3 show that as model capacity and CFG strength increase, the performance differences among solvers, dynamical formulations, and between SDE and ODE sampling become increasingly negligible,a trend that also holds for TADA. Conversely, the results underscore TADA's strength in under-parameterized settings, where modest-capacity DM must tackle large scale, high-dimensional data such as those encountered in video generation. Lastly, TADA currently uses a basic exponential integrator. Evaluating further improvement with more advanced solvers, such as UniPC and DPM-Solver++, remains an critical direction for future work.

**Broader Impacts:** TADA boosts DM efficiency, accelerating the spread of generated content.

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

# Appendix

## A   Notations

In this appendix, we follow the notation introduced before. However, in order to reduce the complexity of the derivation, we simplify all the coefficient with Kronecker product. All the variable which sololy depends on the time, will be broadcast by the kroneker product, For example $\mathbf{A}_t := \mathbf{A}_t \otimes \mathbf{I}_d$, $\boldsymbol{\mu}_t := \boldsymbol{\mu}_t \otimes \mathbf{I}_d$, $\mathbf{L}_t^v v = \mathbf{L}_t^v v \otimes \mathbf{I}_d$ etc.

## B   Proof of Proposition. 3.1

*Proof.* We first analyze the objective function of momentum diffusion model for $N = 2$ case, and it can be generalize to larger $N$. **For clarity in dimensional alignment and derivational correctness, we refer the reader to the simplified notation in Section A, which will also be adopted throughout this section.**

$$\min_\theta \mathcal{L}_{\text{MDM}}(\theta) = \mathbb{E}_{x_1,\epsilon,t} \| \epsilon_\theta(\mathbf{x}_t, t) - \epsilon^{(1)} \|_2^2 \tag{6}$$

$$= \mathbb{E}_{x_1,\epsilon,t} \frac{1}{L_t^{vv2}} \| L_t^{vv} \epsilon_\theta(\mathbf{x}_t, t) - L_t^{vv} \epsilon^{(1)} \|_2^2 \tag{7}$$

$$= \mathbb{E}_{x_1,\epsilon,t} \frac{1}{L_t^{vv2}} \| L_t^{vv} \epsilon_\theta(\mathbf{x}_t, t) - \left[ x_t^{(1)} - \frac{L_t^{xv}}{L_t^{xx}} x_t^{(0)} - \left( \boldsymbol{\mu}_t^{(0)} - \frac{L_t^{xv}}{L_t^{xx}} \boldsymbol{\mu}_t^{(1)} \right) x_1 \right] \|_2^2 \tag{8}$$

$$= \mathbb{E}_{x_1,\epsilon,t} \frac{1}{L_t^{vv2}} \| L_t^{vv} \epsilon_\theta(\mathbf{x}_t, t) - x_t^{(1)} + \frac{L_t^{xv}}{L_t^{xx}} x_t^{(0)} - \left( \frac{L_t^{xv}}{L_t^{xx}} \boldsymbol{\mu}_t^{(1)} - \boldsymbol{\mu}_t^{(0)} \right) x_1 \|_2^2 \tag{9}$$

$$= \mathbb{E}_{x_1,\epsilon,t} \frac{\left( \frac{L_t^{xv}}{L_t^{xx}} \boldsymbol{\mu}_t^{(1)} - \boldsymbol{\mu}_t^{(0)} \right)^2}{L_t^{vv2}} \| \underbrace{\frac{L_t^{vv} \epsilon_\theta(\mathbf{x}_t, t) - x_t^{(1)} + \frac{L_t^{xv}}{L_t^{xx}} x_t^{(0)}}{\frac{L_t^{xv}}{L_t^{xx}} \boldsymbol{\mu}_t^{(1)} - \boldsymbol{\mu}_t^{(0)}}}_{\text{parameterized Neural Netowrk}} - x_1 \|_2^2 \tag{10}$$

Following the same spirit, one can derive the case for $N$ variable. See Appendix. J for details.

We know that,

$$\mathbf{x}_t \mid x_1 \sim \mathcal{N}(\boldsymbol{\mu}_t x_1, \ \boldsymbol{\Sigma}_t),$$

Define

$$\mathbf{r}_t := \frac{\boldsymbol{\Sigma}_t^{-1} \boldsymbol{\mu}_t}{\boldsymbol{\mu}_t^\top \boldsymbol{\Sigma}_t^{-1} \boldsymbol{\mu}_t}, \quad y_t := \mathbf{r}_t^\top \mathbf{x}_t,$$

and the residual ("noise")

$$\boldsymbol{\epsilon} := \mathbf{x}_t - y_t \, \boldsymbol{\mu}_t.$$

Since the operation is linear, one can transform the $\mathbf{x}_t$ by

$$T: \ \mathbf{x}_t \ \mapsto \ (y_t, \boldsymbol{\epsilon}) = \left( \mathbf{r}_t^\top \mathbf{x}_t, \ \mathbf{x}_t - \frac{(\mathbf{r}_t^\top \mathbf{x}_t)}{a_t} \boldsymbol{\mu}_t \right)$$

and it is *invertible* with linear inverse $\mathbf{x}_t = y_t \boldsymbol{\mu}_t + \boldsymbol{\epsilon}$. Hence the $\sigma$-algebras coincide:

$$\sigma(\mathbf{x}_t) = \sigma(y_t, \boldsymbol{\epsilon}).$$

For any integrable random variable $Z$, equal $\sigma$-fields imply $\mathbb{E}[Z \mid \mathbf{x}_t] = \mathbb{E}[Z \mid y_t, \boldsymbol{\epsilon}]$. Taking $Z = x_1$ yields

$$\mathbb{E}[x_1 \mid \mathbf{x}_t] = \mathbb{E}[x_1 \mid y_t, \boldsymbol{\epsilon}].$$

due to the fact that $\boldsymbol{\epsilon}$ is the independent gaussian, thus

$$\mathbb{E}[x_1 \mid \mathbf{x}_t] = \mathbb{E}[x_1 \mid y_t, \boldsymbol{\epsilon}] = \mathbb{E}[x_1 \mid y_t, \boldsymbol{\epsilon}] = \mathbb{E}[x_1 \mid y_t].$$

$\square$

# C Proof of Proposition. 3.3

**For clarity in dimensional alignment and derivational correctness, we refer the reader to the simplified notation in Section A, which will also be adopted throughout this section.**

*Proof.* The dynamics we considered reads

$$\frac{d\mathbf{x}_t}{dt} = \mathbf{A}_t\,\mathbf{x}_t + \mathbf{b}_t F_t, \quad x_0 \sim \mathcal{N}(0, I). \tag{11}$$

and again, in this paper, we only consider,

$$\mathbf{A}_t = \begin{bmatrix} 0 & 1 & 0 & \cdots & 0 \\ 0 & 0 & 1 & \cdots & 0 \\ \vdots & \vdots & \ddots & \ddots & \vdots \\ 0 & 0 & \cdots & 0 & 1 \\ 0 & 0 & \cdots & 0 & 0 \end{bmatrix}_{N \times N}, \text{ and } \quad \mathbf{b}_t := \begin{bmatrix} 0 \\ 0 \\ \vdots \\ \vdots \\ 1 \end{bmatrix} \tag{12}$$

If we expand the system, it basically represent:

$$dx_t^{(0)} = x_t^{(1)} dt \tag{13}$$

$$dx_t^{(1)} = x_t^{(2)} dt \tag{14}$$

$$\cdots \tag{15}$$

$$dx_t^{(N-1)} = F_t dt \tag{16}$$

$$\tag{17}$$

In our case, the $F_t$ function is:

$$F_t := N! \frac{x_1 - \sum_{i=0}^{N-1} \frac{x_t^{(i)}}{i!}(1-t)^i}{(1-t)^N} \tag{18}$$

One can easily verify that, when $N = 1$, it actually degenerate to flow matching:

$$F_t := \frac{x_1 - \mathbf{x}_t}{1 - t}, \text{ and} \tag{19}$$

$$dx_t^{(0)} = F_t dt \tag{20}$$

One can consider it as the higher augment dimension extension of flow matching model. And the magical part is that, we do not need to retrain model.

For better analysis, we rearrange the system:

$$\frac{d\mathbf{x}_t}{dt} = \mathbf{A}_t\,\mathbf{x}_t + \mathbf{b}_t F_t \tag{21}$$

$$= \mathbf{A}_t\,\mathbf{x}_t + \mathbf{b}_t N! \frac{x_1 - \sum_{i=0}^{N-1} \frac{x_t^{(i)}}{i!}(1-t)^i}{(1-t)^N} \tag{22}$$

$$= \mathbf{A}_t\,\mathbf{x}_t + \frac{\mathbf{b}_t N!}{(1-t)^N} x_1 - \frac{\mathbf{b}_t N! \sum_{i=0}^{N-1} \frac{x_t^{(i)}}{i!}(1-t)^i}{(1-t)^N} \tag{23}$$

$$= \mathbf{A}_t\,\mathbf{x}_t + \underbrace{\begin{bmatrix} 0 \\ 0 \\ \vdots \\ \frac{N!}{(1-t)^N} \end{bmatrix}}_{\hat{\mathbf{b}}_t} x_1 - \underbrace{\begin{bmatrix} 0 & 0 & \cdots & 0 \\ 0 & 0 & \cdots & 0 \\ \vdots & \vdots & \ddots & \vdots \\ \frac{(1-t)^0 N!}{0!(1-t)^N} & \frac{(1-t)^1 N!}{1!(1-t)^N} & \cdots & \frac{(1-t)^N N!}{(N-1)!(1-t)^N} \end{bmatrix}}_{\tilde{\mathbf{A}}_t} \mathbf{x}_t \tag{24}$$

$$:= \hat{\mathbf{A}}_t \mathbf{x}_t + \hat{\mathbf{b}}_t x_1, \quad (\hat{\mathbf{A}}_t := \mathbf{A}_t - \tilde{\mathbf{A}}_t) \tag{25}$$

Consider the linear time-varying system:

$$\frac{d\mathbf{x}_t}{dt} = \hat{\mathbf{A}}_t\,\mathbf{x}_t + \hat{\mathbf{b}}_t\,x_1, \quad \mathbf{x}_0 \sim \mathcal{N}(0, I). \tag{26}$$

Since ( 26) is linear and deterministic (apart from the random initial condition), the state remains Gaussian. Its mean and covariance evolve as follows [26].

**Mean Dynamics.**   Let
$$\mathbf{m}_t = \mathbb{E}[\mathbf{x}_t].$$
Then
$$\dot{\mathbf{m}}_t = \hat{\mathbf{A}}_t\,\mathbf{m}_t + \hat{\mathbf{b}}_t\,x_1, \quad m_0 = 0. \tag{27}$$
We can write the mean in a factorized form as
$$\mathbf{m}_t = \boldsymbol{\mu}_t\,x_1,$$
so that by dividing by $x_1$, we obtain
$$\dot{\boldsymbol{\mu}}_t = \hat{\mathbf{A}}_t\,\boldsymbol{\mu}_t + \hat{\mathbf{b}}_t.$$

**Covariance Dynamics.**   Similarly, the convariance follows dynamics
$$\dot{\boldsymbol{\Sigma}}_t = \hat{\mathbf{A}}_t\,\boldsymbol{\Sigma}_t + \boldsymbol{\Sigma}_t\,\hat{\mathbf{A}}_t^T \tag{28}$$
Recall that, Then we define the scalar quantity of interest as

$$y_t := \frac{\left(\boldsymbol{\Sigma}_t^{-1}\mathbf{m}_t\right)^T \mathbf{x}_t}{\left(\boldsymbol{\Sigma}_t^{-1}\mathbf{m}_t\right)^T \boldsymbol{\mu}_t} = \frac{\left(\boldsymbol{\Sigma}_t^{-1}\boldsymbol{\mu}_t\right)^T \mathbf{x}_t}{\boldsymbol{\mu}_t^T \boldsymbol{\Sigma}_t^{-1}\boldsymbol{\mu}_t} = \frac{\mathbf{r}_t^T \mathbf{x}_t}{\gamma_t}, \quad \text{with } \gamma_t := \boldsymbol{\mu}_t^T \boldsymbol{\Sigma}_t^{-1}\boldsymbol{\mu}_t. \tag{29}$$

We wish to compute the derivative of
$$y_t = \frac{\mathbf{r}_t^T \mathbf{x}_t}{\gamma_t},$$
Using the quotient rule,
$$\dot{y}_t = \frac{\frac{d}{dt}\left(\mathbf{r}_t^T \mathbf{x}_t\right)\gamma_t - \left(\mathbf{r}_t^T \mathbf{x}_t\right)\dot{\gamma}_t}{\gamma_t^2}.$$
Since $\mathbf{r}_t^T \mathbf{x}_t = y_t \gamma_t$, this becomes
$$\dot{y}_t = \frac{\dot{\mathbf{r}}_t^T \mathbf{x}_t + \mathbf{r}_t^T \dot{x}_t}{\gamma_t} - y_t \frac{\dot{\gamma}_t}{\gamma_t}.$$

**Derivative of $\mathbf{r}_t$**

Recall that
$$\mathbf{r}_t = \boldsymbol{\Sigma}_t^{-1}\boldsymbol{\mu}_t.$$
Differentiating gives
$$\dot{\mathbf{r}}_t = -\boldsymbol{\Sigma}_t^{-1}\dot{\boldsymbol{\Sigma}}_t\,\boldsymbol{\Sigma}_t^{-1}\boldsymbol{\mu}_t + \boldsymbol{\Sigma}_t^{-1}\dot{\boldsymbol{\mu}}_t.$$
Substitute the known dynamics:
$$\dot{\boldsymbol{\Sigma}}_t = \hat{\mathbf{A}}_t\,\boldsymbol{\Sigma}_t + \boldsymbol{\Sigma}_t\,\hat{\mathbf{A}}_t^T,$$
$$\dot{\boldsymbol{\mu}}_t = \hat{\mathbf{A}}_t\,\boldsymbol{\mu}_t + \hat{\mathbf{b}}_t.$$
It follows that

$$\dot{\mathbf{r}}_t = -\boldsymbol{\Sigma}_t^{-1}\dot{\boldsymbol{\Sigma}}_t\,\boldsymbol{\Sigma}_t^{-1}\boldsymbol{\mu}_t + \boldsymbol{\Sigma}_t^{-1}\dot{\boldsymbol{\mu}}_t \tag{30}$$
$$= -\boldsymbol{\Sigma}_t^{-1}\left(\hat{\mathbf{A}}_t\,\boldsymbol{\Sigma}_t + \boldsymbol{\Sigma}_t\,\hat{\mathbf{A}}_t^T\right)\boldsymbol{\Sigma}_t^{-1}\boldsymbol{\mu}_t + \boldsymbol{\Sigma}_t^{-1}\left(\hat{\mathbf{A}}_t\,\boldsymbol{\mu}_t + \hat{\mathbf{b}}_t\right) \tag{31}$$
$$= -\boldsymbol{\Sigma}_t^{-1}\hat{\mathbf{A}}_t\boldsymbol{\mu}_t - \hat{\mathbf{A}}_t^T\boldsymbol{\Sigma}_t^{-1}\boldsymbol{\mu}_t + \boldsymbol{\Sigma}_t^{-1}\hat{\mathbf{A}}_t\boldsymbol{\mu}_t + \boldsymbol{\Sigma}_t^{-1}\hat{\mathbf{b}}_t \tag{32}$$
$$= -\hat{\mathbf{A}}_t^T\mathbf{r}_t + \boldsymbol{\Sigma}_t^{-1}\hat{\mathbf{b}}_t. \tag{33}$$

**Derivative of $\mathbf{x}_t$**

From ( 26),

$$\dot{\mathbf{x}}_t = \hat{\mathbf{A}}_t\,\mathbf{x}_t + \hat{\mathbf{b}}_t\,\hat{x}_1.$$

**Derivative of $\gamma_t$**

Recall

$$\gamma_t = {\boldsymbol{\mu}_t}^T\,{\boldsymbol{\Sigma}_t}^{-1}\,\boldsymbol{\mu}_t = \mathbf{r}_t^T\,\boldsymbol{\mu}_t.$$

Differentiating,

$$\dot{\gamma}_t = \dot{\mathbf{r}}_t^T\,\boldsymbol{\mu}_t + \mathbf{r}_t^T\,\dot{\boldsymbol{\mu}}_t.$$

Using ( 30) and $\dot{\boldsymbol{\mu}}_t = \hat{A}_t\boldsymbol{\mu}_t + \hat{\mathbf{b}}_t$, one obtains (after cancellation) the result:

$$\dot{\gamma}_t = \left(-\hat{\mathbf{A}}_t^\mathsf{T}\mathbf{r}_t + {\boldsymbol{\Sigma}_t}^{-1}\hat{\mathbf{b}}_t\right)\boldsymbol{\mu}_t + \mathbf{r}^\mathsf{T}\left(\hat{\mathbf{A}}_t\boldsymbol{\mu}_t + \hat{\mathbf{b}}_t\right) \tag{34}$$

$$= 2\,\hat{\mathbf{b}}_t^T\,\mathbf{r}_t \tag{35}$$

$$= 2\,\hat{\mathbf{b}}_t^T\,{\boldsymbol{\Sigma}_t}^{-1}\boldsymbol{\mu}_t \tag{36}$$

**Combining Everything**

Substitute the pieces into

$$\dot{y}_t = \frac{\dot{\mathbf{r}}_t^T\,\mathbf{x}_t + \mathbf{r}_t^T\,\dot{\mathbf{x}}_t}{\gamma_t} - y_t\,\frac{\dot{\gamma}_t}{\gamma_t}.$$

Using

$$\dot{\mathbf{r}}_t^T = -\mathbf{r}_t^T\hat{\mathbf{A}}_t + \hat{\mathbf{b}}_t^T\,{\boldsymbol{\Sigma}_t}^{-1},$$

$$\mathbf{r}_t^T\dot{\mathbf{x}}_t = \mathbf{r}_t^T\left(\hat{\mathbf{A}}_t\mathbf{x}_t + \hat{\mathbf{b}}_t\,\hat{x}_1\right),$$

we have:

$$\dot{\mathbf{r}}_t^T\,\mathbf{x}_t + \mathbf{r}_t^T\,\dot{\mathbf{x}}_t = \left[-\mathbf{r}_t^T\hat{\mathbf{A}}_t\,\mathbf{x}_t + \hat{\mathbf{b}}_t^T\,{\boldsymbol{\Sigma}_t}^{-1}\,\mathbf{x}_t\right] + \left[\mathbf{r}_t^T\hat{\mathbf{A}}_t\,\mathbf{x}_t + \mathbf{r}_t^T\,\hat{\mathbf{b}}_t\,\hat{x}_1\right]$$

$$= \hat{\mathbf{b}}_t^T\,{\boldsymbol{\Sigma}_t}^{-1}\,\mathbf{x}_t + \mathbf{r}_t^T\,\hat{\mathbf{b}}_t x_1$$

Thus,

$$\dot{y}_t = \frac{\hat{\mathbf{b}}_t^T\,{\boldsymbol{\Sigma}_t}^{-1}\,\mathbf{x}_t + x_1\,\mathbf{r}_t^T\,\hat{\mathbf{b}}_t}{\gamma_t} - y_t\,\frac{\dot{\gamma}_t}{\gamma_t}. \tag{37}$$

Recall that $\gamma_t = {\boldsymbol{\mu}_t}^T\,{\boldsymbol{\Sigma}_t}^{-1}\,\boldsymbol{\mu}_t$ and $\dot{\gamma}_t = 2\,\hat{\mathbf{b}}_t^T\,{\boldsymbol{\Sigma}_t}^{-1}\,\boldsymbol{\mu}_t$. Also note that

$$\mathbf{r}_t^T\,\hat{\mathbf{b}}_t = {\boldsymbol{\mu}_t}^T\,{\boldsymbol{\Sigma}_t}^{-1}\,\hat{\mathbf{b}}_t.$$

Thus, the final expression becomes

$$\dot{y}_t = \frac{\hat{\mathbf{b}}_t^T\,{\boldsymbol{\Sigma}_t}^{-1}\,\mathbf{x}_t + \hat{x}_1\,{\boldsymbol{\mu}_t}^T\,{\boldsymbol{\Sigma}_t}^{-1}\,\hat{\mathbf{b}}_t}{{\boldsymbol{\mu}_t}^T\,{\boldsymbol{\Sigma}_t}^{-1}\,\boldsymbol{\mu}_t} - y_t\,\frac{2\,\hat{\mathbf{b}}_t^T\,{\boldsymbol{\Sigma}_t}^{-1}\,\boldsymbol{\mu}_t}{{\boldsymbol{\mu}_t}^T\,{\boldsymbol{\Sigma}_t}^{-1}\,\boldsymbol{\mu}_t} \tag{38}$$

$$= \underbrace{\frac{\hat{\mathbf{b}}_t^T\,{\boldsymbol{\Sigma}_t}^{-1}}{{\boldsymbol{\mu}_t}^T\,{\boldsymbol{\Sigma}_t}^{-1}\,\boldsymbol{\mu}_t}}_{\mathbf{e}_t}\mathbf{x}_t + \frac{{\boldsymbol{\mu}_t}^T\,{\boldsymbol{\Sigma}_t}^{-1}\,\hat{\mathbf{b}}_t}{{\boldsymbol{\mu}_t}^T\,{\boldsymbol{\Sigma}_t}^{-1}\,\boldsymbol{\mu}_t}x_1 - \frac{2\,\hat{\mathbf{b}}_t^T\,{\boldsymbol{\Sigma}_t}^{-1}\,\boldsymbol{\mu}_t}{{\boldsymbol{\mu}_t}^T\,{\boldsymbol{\Sigma}_t}^{-1}\,\boldsymbol{\mu}_t}y_t \tag{39}$$

The frist term is essentially one kind of linear combination of $\mathbf{x}_t$, and Recally that $y_t = \mathbf{r}^\mathsf{T}\mathbf{x}_t :=$ $\frac{\boldsymbol{\mu}_t^\mathsf{T}\boldsymbol{\Sigma}_t^{-1}}{\boldsymbol{\mu}_t^\mathsf{T}\boldsymbol{\Sigma}_t^{-1}\boldsymbol{\mu}_t}$ which is another linear combination of $\mathbf{x}_t$. Assume that $\mathbf{x}_t \sim \mathcal{N}(\boldsymbol{\mu}_t x_1, \boldsymbol{\Sigma}_t)$, thus, one can derive the relationship between the frist term and $y_t$. Thus, according to Lemma. I.1

$$\frac{\hat{\mathbf{b}}_t^T\,{\boldsymbol{\Sigma}_t}^{-1}}{{\boldsymbol{\mu}_t}^T\,{\boldsymbol{\Sigma}_t}^{-1}\,\boldsymbol{\mu}_t}\mathbf{x}_t = \mathbf{e}_t^\mathsf{T}\left[\mathbf{I} - \frac{\boldsymbol{\Sigma}_t\mathbf{r}_t\mathbf{r}_t^\mathsf{T}}{\mathbf{r}_t^\mathsf{T}\boldsymbol{\Sigma}_t\mathbf{r}_t}\right]\boldsymbol{\mu}_t x_1 + \frac{\mathbf{e}_t^\mathsf{T}\boldsymbol{\Sigma}_t\mathbf{r}_t}{\mathbf{r}_t^\mathsf{T}\boldsymbol{\Sigma}_t\mathbf{r}_t}y_t + \mathbf{e}_t^\mathsf{T}\mathbf{L}_t\boldsymbol{\epsilon}_\perp, \tag{40}$$

$$\boldsymbol{\epsilon}_\perp \sim \mathcal{N}\left(\mathbf{0},\ I_d - \mathbf{L}_t^\mathsf{T}\mathbf{r}_t\mathbf{r}_t^\mathsf{T}\mathbf{L}_t/\mathbf{r}_t^\mathsf{T}\boldsymbol{\Sigma}_t\mathbf{r}_t\right). \tag{41}$$

Thus, by plugging in the expression, one can get:

$$\dot{y}_t = \alpha_t y_t + \beta x_1 + \mathbf{e}_t^\mathsf{T} \mathbf{L}_t \boldsymbol{\epsilon}_\perp \tag{42}$$

$$\approx \alpha_t y_t + \beta x_\theta + \mathbf{e}_t^\mathsf{T} \mathbf{L}_t \boldsymbol{\epsilon}_\perp \tag{43}$$

Where,

$$\alpha_t = \frac{\mathbf{e}_t^\mathsf{T} \boldsymbol{\Sigma}_t \mathbf{r}_t}{\mathbf{r}_t^\mathsf{T} \boldsymbol{\Sigma}_t \mathbf{r}_t} - \frac{2\, \hat{\mathbf{b}}_t^T \, \boldsymbol{\Sigma}_t^{-1} \, \boldsymbol{\mu}_t}{\boldsymbol{\mu}_t^T \, \boldsymbol{\Sigma}_t^{-1} \, \boldsymbol{\mu}_t} \tag{44}$$

$$\beta_t = \frac{\boldsymbol{\mu}_t^T \, \boldsymbol{\Sigma}_t^{-1} \, \hat{\mathbf{b}}_t}{\boldsymbol{\mu}_t^T \, \boldsymbol{\Sigma}_t^{-1} \, \boldsymbol{\mu}_t} + \mathbf{e}_t^\mathsf{T} \left[ \mathbf{I} - \frac{\boldsymbol{\Sigma}_t \mathbf{r}_t \mathbf{r}_t^\mathsf{T}}{\mathbf{r}_t^\mathsf{T} \boldsymbol{\Sigma}_t \mathbf{r}_t} \right] \boldsymbol{\mu}_t \tag{45}$$

$$w_t^{(i)} = (\mathbf{e}_t^\mathsf{T} \mathbf{L}_t)^{(i)} \tag{46}$$

$\square$

## D  Experiment Details

Here we elaborate more on experiment details.

### D.1  EDM and EDM2

For the baselines on EDM and EDM2 codebase, we directly use the code provide in DPM-Solver-v3 [36]. For the fair comparision, for all baselines, we controlled $\sigma_{\min} = 0.002$ and $\sigma_{\max} = 80$ as suggested in the original EDM and EDM2 paper.

For DPM-Solver++, we did abalation search over order $\in [1, 2, 3]$, discretization $\in$ [logSNR, time uniform, , edm, time quadratic].

For UniPC, we did abalation search over order $\in [1, 2, 3]$, discretization $\in$ [logSNR, time uniform, , edm, time quadratic], variant $\in$ [bh1, bh2].

For all the ablation results, please see the supplementary material.

### D.2  Stable Diffusion 3

For stable diffusion 3, we simply plug in the implementation of all the baselines provided in the Diffuser. We use latest HpsV2.1 to evaluate generate dresults.

## E  Additional Plots

### E.1  General $N$ variable dynamics

This section is not referenced in the main paper and will be removed soon; it is retained only for now to keep the appendix numbering aligned with the main paper.

## F  Detailed Explanations

### F.1  Explicit form of $\mathbf{A}_t$ and $\mathbf{b}_t$

Here we demonstrate the $\mathbf{A}_t$ and $\mathbf{b}_t$ used in AGM [6] and CLD [9]. Here we abuse the notation and inherent the notation from CLD.

### F.2  Explicit form of mean and convariance matrix

Here we first quickly derive how the $F$ derived which is straight-forward. We know $x_t^{(0)} := x_t$ be the position and define higher derivatives recursively

$$x_t^{(k)} = \frac{d^k x_t^{(0)}}{dt^k}, \quad k = 1, \dots, N - 1.$$

Table 1: Comparison of different solvers

| Algorithm | $\mathbf{A}_t$ | $\mathbf{b}$ | $F_t$ |
|-----------|----------------|--------------|-------|
| AGM [9] | $\begin{bmatrix} 0 & 1 \\ 0 & 0 \end{bmatrix}$ | $[0,1]^\mathsf{T}$ | $\frac{-4}{t-1}\left(\frac{x1-\mathbf{x}_t^{(0)}}{1-t} - \mathbf{x}_t^{(1)}\right)$ |
| CLD [9] | $\begin{bmatrix} 0 & -M^{-1} \\ 1 & \Gamma M^{-1} \end{bmatrix}\beta$ | $[0,\Gamma\beta]^\mathsf{T}$ | $\nabla_{\mathbf{x}_t^{(1)}} \log p(\mathbf{x},t)$ |

The system dynamics form an *Nth–order chain of integrators* driven by a scalar input $F(t,\mathbf{x}_t)$:

$$\dot{x}_t^{(0)} = x_t^{(1)},$$
$$\dot{x}_t^{(1)} = x_t^{(2)},$$
$$\vdots$$
$$\dot{x}_t^{(N-2)} = x_t^{(N-1)},$$
$$\dot{x}_t^{(N-1)} = F(t,\mathbf{x}_t). \tag{47}$$

Equivalently, the position satisfies the scalar ODE

$$\boxed{\frac{d^N x_t^{(0)}}{dt^N} = F(t,\mathbf{x}_t)}.$$

Our goal is starting at some time $t \in [0,1)$ with known state $\{x_t^{(k)}\}_{k=0}^{N-1}$, choose $F$ so that the position reaches a prescribed value at $t=1$:

$$x_1^{(0)} = x_1 \quad \text{(``hit the target'')}.$$

Assume $F$ is *held constant* over the *remaining* interval $[t,1]$. Repeated integration yields the degree-$N$ Taylor polynomial about $t$:

$$x_1^{(0)} = \sum_{k=0}^{N-1} \frac{(1-t)^k}{k!} x_t^{(k)} + \frac{(1-t)^N}{N!} F. \tag{48}$$

Thus, one can simply solve the F by Rearranging (48) to isolate $F$:

$$F = \frac{N!}{(1-t)^N}\left[x_1 - \sum_{k=0}^{N-1} \frac{(1-t)^k}{k!} x_t^{(k)}\right].$$

Thanks to the simple form of $F$, one can readily write down the mean and covariance of the system.

Now we know

$$A = \begin{bmatrix} 0 & 1 & 0 & \dots & 0 \\ 0 & 0 & 1 & \dots & 0 \\ \vdots & \vdots & \ddots & \ddots & \vdots \\ 0 & 0 & \dots & 0 & 1 \\ 0 & 0 & \dots & 0 & 0 \end{bmatrix}, \qquad \mathbf{b}_t = [0,\dots,0,1]^\mathsf{T}.$$

By rearraging the dynamics, gives the *linear* time-varying closed loop

$$\dot{\mathbf{x}}_t = \underbrace{\hat{\mathbf{A}}_t}_{A \text{ w/ control}} \mathbf{x}_t + \underbrace{\hat{\mathbf{b}}}_{b \text{ w/ control}} x_1 \tag{49}$$

We need first compute the transition matrix induced by $\hat{\mathbf{A}}_t$ and we call it controlled transition matrix. By Solving $\dot{\Phi} = \hat{\mathbf{A}}_t \Phi$ column-wise gives the polynomial matrix

$$\Phi(t,0) = \left[T_{k,m}(t)\right]_{k,m=0}^{N-1}, \quad T_{k,m}(t) = \begin{cases} \dfrac{t^{m-k}}{(m-k)!} - \dfrac{N!\,t^{N-k}}{(N-k)!\,m!}, & m \geq k, \\ -\dfrac{N!\,t^{N-k}}{(N-k)!\,m!}, & m < k. \end{cases}$$

Plugging this $\Phi(t,0)$ into the boxed formulas above supplies $\mu(t)$ and $\Sigma(t)$ explicitly for *every* order $N$.

Let $\mu(t) = \mathbb{E}[\mathbf{x}_t]$. Because $\hat{\mathbf{b}}_t x_1$ is deterministic,

$$\dot{\mu}(t) = \hat{\mathbf{A}}_t\,\mu(t) + \hat{\mathbf{b}}_t\,x_1, \qquad \mu(0) = \mu_0. \tag{50}$$

Define the state–transition matrix $\Phi(t,\tau)$ of $\hat{\mathbf{A}}_t$:

$$\dot{\Phi}(t,\tau) = \hat{\mathbf{A}}_t\,\Phi(t,\tau), \ \ \Phi(\tau,\tau) = I.$$

Then the standard variation-of-constants formula gives

$$\boldsymbol{\mu}_t = \Phi(t,0)\,\boldsymbol{\mu}_0 + \int_0^t \Phi(t,\tau)\,\hat{\mathbf{b}}_\tau\,x_1\,d\tau.$$

If the initial derivatives are i.i.d. $\mathcal{N}(0,1)$, then $\mu_0 = \mathbf{0}$ and only the integral term remains. Carrying out the integral (polynomials of $\tau$) yields

$$\boxed{\boldsymbol{\mu}_t^{(k)} = \frac{N!\,t^{N-k}}{(N-k)!}\,x_1, \qquad k = 0, \ldots, N-1.}$$

Meanwhile, the propagation of covariance matrix is:

$$\dot{\boldsymbol{\Sigma}}_t = \hat{\mathbf{A}}_t\boldsymbol{\Sigma}_t + \boldsymbol{\Sigma}_t\hat{\mathbf{A}}_t^\top, \quad \Sigma(0) = \Sigma_0. \tag{51}$$

Eq. 51 is a homogeneous Lyapunov ODE whose unique solution is exactly (see Appendix. H for more details):

$$\boxed{\boldsymbol{\Sigma}_t = \Phi(t,0)\,\boldsymbol{\Sigma}_0\,\Phi(t,0)^\top.}$$

## F.3 Previous Fast Solver

Table 2: Comparison of different solvers

|  | Order type | Order | Multistep type | Expansion term | Discretize space |
|---|---|---|---|---|---|
| Heun [17] | Single Step | 2 | N/A | N/A | $\sigma_t$ |
| DEIS [33] | Multi-step | 2/3/4 | Adams–Bashforth | $\epsilon_\theta$ | $\sigma_t$ |
| DPM-Solver [22] | Multi/Single-step | 2/3/4 | Adams–Bashforth | $\epsilon_\theta$ | Optional |
| DPM-Solver++ [23] | Multi/Single-step | 2/3/4 | Adams–Bashforth | $x_\theta$ | Optional |
| UniPC [34] | Multi-step | 3/4/5 | Adams–Moulton | $x_\theta$ | Optional |
| TADA(ours) | Multi-step | 2/3 | Adams–Bashforth | $F_\theta$ | $t$ |

## F.4 Extended Flow Matching

In the framework of flow mathcing, one obtain the velocity by $v_t = \frac{x_1 - x_t}{1-t}$ because it is the linear interpolation between data $x_1$ and prior $x_0$. And meanwhile, it happens to be the solution of optimal control problem:

$$\min_{v_t} \int_t^1 \|v_t\|_2^2 dt, \quad s.t \quad \mathrm{d}x_t = v_t \mathrm{d}t \tag{52}$$

For the detailed derivation, please see Sec.C.1 in [6].

For AGM, they consider a momentum system, which reads

$$\min_{a_t} \int_t^1 \|a_t\|_2^2 \mathrm{d}t, \quad s.t \quad \mathrm{d}x_t = v_t \mathrm{d}t, \quad \mathrm{d}v_t = a_t \mathrm{d}t + \mathrm{d}w_t \tag{53}$$

The differences is that, AGM consider the injection of stochasticity in the velocity channel. For our case, the $F_\theta$ derived in Sec. 3.2, is the solution for

$$\min_{F_t} \int_t^1 \|F_t\|_2^2 \mathrm{d}t$$
$$\mathrm{d}x_t^{(0)} = x_t^{(1)} \mathrm{d}t$$
$$\mathrm{d}x_t^{(1)} = x_t^{(2)} \mathrm{d}t$$
$$\cdots$$
$$\mathrm{d}x_t^{(N-1)} = F_t \mathrm{d}t$$

and its spirit keeps same as previous formulation, move $x_t^{(0)}$ to $x_1^0 \sim p_{\text{data}}$ from $t = 0$ to $t = 1$.

### F.5 Degenerate Case of TADA

Here we discuss about the degenerated case of TADA. The reasoning behind it is rather simple. We show the dynamics of $y_t$ (eq. 39) again here,

$$\dot{y}_t = \underbrace{\frac{\hat{\mathbf{b}}_t^T \mathbf{\Sigma}_t^{-1}}{\boldsymbol{\mu}_t^T \mathbf{\Sigma}_t^{-1} \boldsymbol{\mu}_t}}_{\mathbf{e}_t} \mathbf{x}_t + \frac{\boldsymbol{\mu}_t^T \mathbf{\Sigma}_t^{-1} \hat{\mathbf{b}}_t}{\boldsymbol{\mu}_t^T \mathbf{\Sigma}_t^{-1} \boldsymbol{\mu}_t} x_1 - \frac{2 \hat{\mathbf{b}}_t^T \mathbf{\Sigma}_t^{-1} \boldsymbol{\mu}_t}{\boldsymbol{\mu}_t^T \mathbf{\Sigma}_t^{-1} \boldsymbol{\mu}_t} y_t \tag{54}$$

and recall that

$$y_t = \frac{\boldsymbol{\mu}_t^\mathsf{T} \mathbf{\Sigma}_t^{-1}}{\boldsymbol{\mu}_t^\mathsf{T} \mathbf{\Sigma}_t^{-1} \boldsymbol{\mu}_t} \tag{55}$$

Thus, if the first term depends exclusively on $y_t$, the system reduces to the scalar ODE for $y_t$ and becomes formally identical to other diffusion-model parameterizations such as VP, VE, or FM. More precisely, in order to degenerate TADA, one only requires

$$\boldsymbol{\mu}_t \; \propto \; \hat{\mathbf{b}}_t \tag{56}$$

where $\hat{\mathbf{b}}_t$ is defined in eq. 24. This proportionality holds in two scenarios:

1. When $N = 1$, so that $\boldsymbol{\mu}_t$ and $\hat{\mathbf{b}}_t$ are scalars. In that case, the framework collapses to flow matching—a mere reparameterization of the diffusion model.

2. When $\mathbf{A}_t$ is diagonal and its components evolve independently. Then every dimension of $\boldsymbol{\mu}_t$ and $\hat{\mathbf{b}}_t$ shares the same mean and variance, and proportionality follows directly.

A simple empirical check is to propagate the model from different random initializations using our formulation: it yields identical FID scores after generation, confirming the degeneracy.

## G  Additional Qualitative Comparision

Please see fig. 10 and fig. 9.

## H  Solution of the homogeneous Lyapunov ODE

Let $\Phi(t, \tau)$ be the *state–transition matrix* of the (possibly time–varying) coefficient $A_t$:

$$\dot{\Phi}(t, \tau) = A_t \, \Phi(t, \tau), \qquad \Phi(\tau, \tau) = I.$$

Throughout we abbreviate $\Phi(t, 0) \equiv \Phi(t)$.

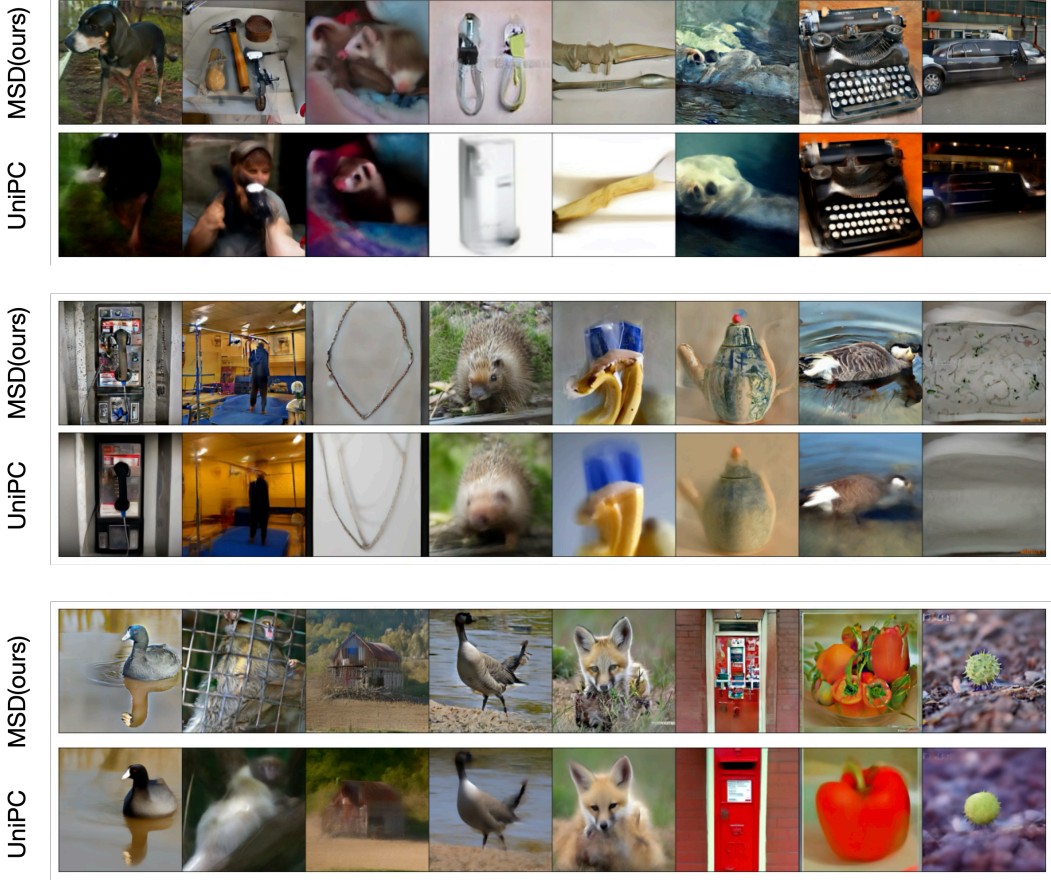

Figure 9: Additional Visual comparision with UniPC using EDM2 w/ 5 NFEs.

**1. Candidate solution.** Consider

$$\Sigma(t) \;=\; \Phi(t)\,\Sigma_0\,\Phi(t)^{\top}.$$

**2. Verification.** Differentiate previous equation and use the product rule together with $\dot{\Phi}(t) = A_t\Phi(t)$ and $\frac{d}{dt}\Phi(t)^{\top} = \Phi(t)^{\top}A_t^{\top}$:

$$\begin{aligned}
\dot{\Sigma}(t) &= \dot{\Phi}\,\Sigma_0\Phi^{\top} + \Phi\,\Sigma_0\dot{\Phi}^{\top} \\
&= A_t\Phi\Sigma_0\Phi^{\top} + \Phi\Sigma_0\Phi^{\top}A_t^{\top} \\
&= A_t\Sigma(t) + \Sigma(t)A_t^{\top}.
\end{aligned}$$

Hence $\Sigma(t)$ satisfies the differential equation in (Lyap), and $\Sigma(0) = \Phi(0)\Sigma_0\Phi(0)^{\top} = \Sigma_0$.

**3. Uniqueness.** Lyapunov Equation is linear in the matrix variable $\Sigma$; by the Picard–Lindelöf theorem its solution is unique. Therefore (S) is *the* solution.

# I  Correlation between two Gaussian Variable

**Lemma I.1.** *Let the random vector*

$$\mathbf{x}_t \;\sim\; \mathcal{N}\big(\boldsymbol{\mu}\,x_1,\; \boldsymbol{\Sigma}_t\big), \qquad \boldsymbol{\Sigma}_t = \mathbf{L}_t\mathbf{L}_t^{\top} \quad (\textit{Cholesky factorisation}).$$

*For two fixed column-vectors* $\mathbf{r}, \mathbf{e} \in \mathbb{R}^d$ *set*

$$y_t := \mathbf{r}_t^{\top}\mathbf{x}_t, \quad z_t := \mathbf{e}_t^{\top}\mathbf{x}_t.$$

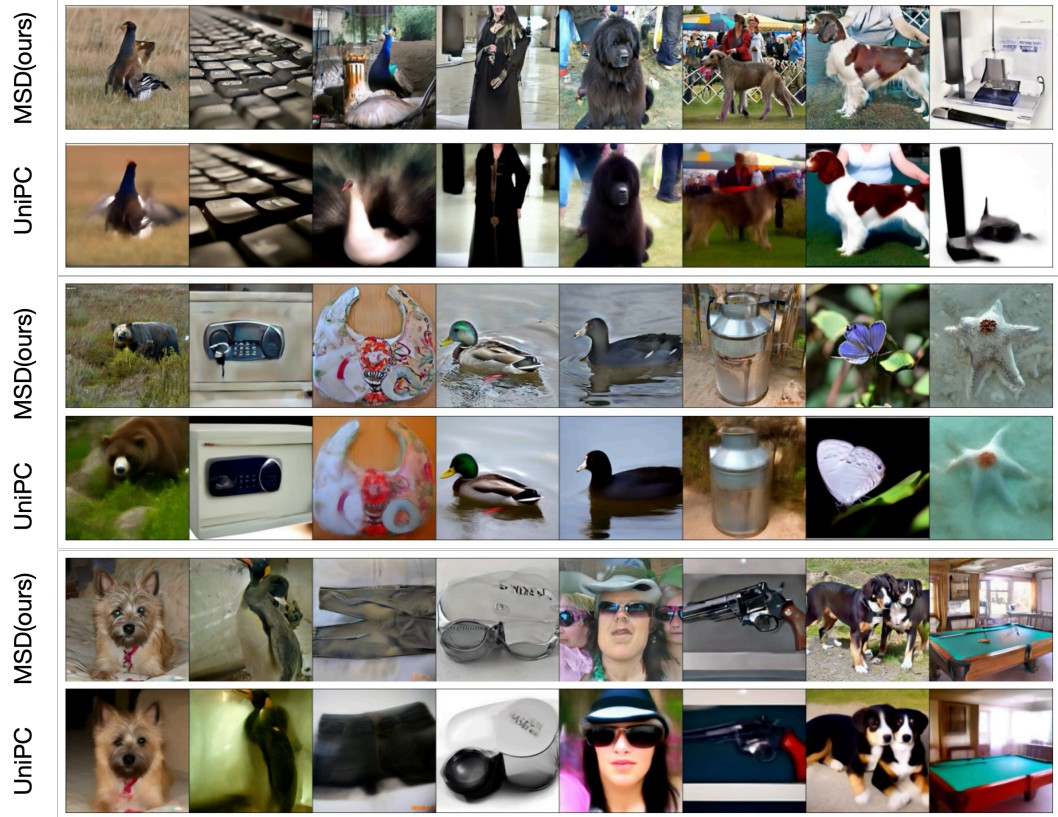

Figure 10: Additional Visual comparision with UniPC using EDM2 w/ 5 NFEs.

*Write the convenient abbreviations*

$$c_t := \mathbf{r}_t^\top \boldsymbol{\mu}_t \, x_1, \quad d_t := \mathbf{e}_t^\top \boldsymbol{\mu}_t \, x_1, \quad \mathbf{g}_t := \mathbf{L}_t^\top \mathbf{r}_t, \quad \mathbf{h}_t := \mathbf{L}_t^\top \mathbf{e}_t, \quad \sigma_y^2 := \|\mathbf{g}_t\|^2.$$

*Then*

$$z_t = \mathbf{e}_t^\top \left[ \mathbf{I} - \frac{\boldsymbol{\Sigma}_t \mathbf{r}_t \mathbf{r}_t^\top}{\mathbf{r}_t^\top \boldsymbol{\Sigma}_t \mathbf{r}_t} \right] \boldsymbol{\mu}_t x_1 + \frac{\mathbf{e}_t^\top \boldsymbol{\Sigma}_t \mathbf{r}_t}{\mathbf{r}_t^\top \boldsymbol{\Sigma}_t \mathbf{r}_t} y_t + \mathbf{e}_t^\top \mathbf{L}_t \boldsymbol{\epsilon}_\perp, \qquad \boldsymbol{\epsilon}_\perp \sim \mathcal{N}\big(\mathbf{0}, \, I_d - \mathbf{L}_t^\top \mathbf{r}_t \mathbf{r}_t^\top \mathbf{L}_t / \mathbf{r}_t^\top \boldsymbol{\Sigma}_t \mathbf{r}_t\big).$$

*Proof.*

$$y_t = c_t + \mathbf{g}_t^\top \boldsymbol{\epsilon}, \qquad z_t = d_t + \mathbf{h}_t^\top \boldsymbol{\epsilon}.$$

Any vector can be decomposed into the component along $\mathbf{s}$ and the component orthogonal to $\mathbf{s}$:

$$\boldsymbol{\epsilon} = \frac{\mathbf{g}_t}{\sigma_y^2} \left( \mathbf{g}_t^\top \boldsymbol{\epsilon} \right) + \boldsymbol{\epsilon}_\perp, \qquad \mathbf{g}_t^\top \boldsymbol{\epsilon}_\perp = 0$$

Because $\boldsymbol{\epsilon} \sim \mathcal{N}(\mathbf{0}, I_d)$ and the projector onto $\mathbf{g}_t$ is orthogonal to the projector onto the complement, $\mathbf{g}_t^\top \boldsymbol{\epsilon}$ and $\boldsymbol{\epsilon}_\perp$ are independent Gaussian variables.

Insert $\mathbf{g}_t^\top \boldsymbol{\epsilon} = y_t - c_t$ to obtain

$$\boldsymbol{\epsilon} = \frac{\mathbf{g}_t}{\sigma_y^2} \left( y_t - c_t \right) + \boldsymbol{\epsilon}_\perp, \qquad \boldsymbol{\epsilon}_\perp \sim \mathcal{N}\big(\mathbf{0}, \, I_d - \mathbf{g}_t \mathbf{g}_t^\top / \sigma_y^2\big).$$

Then we can plug this into $z_t$:

$$z_t = d_t + \mathbf{h}_t^\top \left( \frac{\mathbf{g}_t}{\sigma_y^2} (y_t - c_t) + \boldsymbol{\epsilon}_\perp \right) \tag{57}$$

$$= \mathbf{e}_t^\top \boldsymbol{\mu}_t x_1 + \frac{\mathbf{h}_t^\top \mathbf{g}_t}{\sigma_y^2} (y_t - c_t) + \mathbf{h}_t^\top \boldsymbol{\epsilon}_\perp \tag{58}$$

$$= \mathbf{e}_t^\top \boldsymbol{\mu}_t x_1 + \frac{\mathbf{h}_t^\top \mathbf{g}_t}{\sigma_y^2} y_t - \frac{\mathbf{h}_t^\top \mathbf{g}_t \mathbf{r}_t^\top \boldsymbol{\mu}_t}{\sigma_y^2} x_1 + \mathbf{h}_t^\top \boldsymbol{\epsilon}_\perp \tag{59}$$

$$= \mathbf{e}_t^\top \boldsymbol{\mu}_t x_1 + \frac{\mathbf{e}_t^\top \boldsymbol{\Sigma}_t \mathbf{r}_t}{\mathbf{r}_t^\top \boldsymbol{\Sigma}_t \mathbf{r}_t} y_t - \frac{\mathbf{e}_t^\top \boldsymbol{\Sigma}_t \mathbf{r}_t}{\mathbf{r}_t^\top \boldsymbol{\Sigma}_t \mathbf{r}_t} \left( \mathbf{r}_t^\top \boldsymbol{\mu}_t \right) x_1 + \mathbf{e}_t^\top \mathbf{L}_t \boldsymbol{\epsilon}_\perp \tag{60}$$

$$= \mathbf{e}_t^\top \left[ \mathbf{I} - \frac{\boldsymbol{\Sigma}_t \mathbf{r}_t \mathbf{r}_t^\top}{\mathbf{r}_t^\top \boldsymbol{\Sigma}_t \mathbf{r}_t} \right] \boldsymbol{\mu}_t x_1 + \frac{\mathbf{e}_t^\top \boldsymbol{\Sigma}_t \mathbf{r}_t}{\mathbf{r}_t^\top \boldsymbol{\Sigma}_t \mathbf{r}_t} y_t + \mathbf{e}_t^\top \mathbf{L}_t \boldsymbol{\epsilon}_\perp \tag{61}$$

$\square$

## J  General $N$ variable MDM loss

At time $t$ the $N$-variables are generated by

$$\boxed{\mathbf{x}_t = \boldsymbol{\mu}_t \, x_1 + \mathbf{L}_t \, \boldsymbol{\varepsilon}} \qquad \boldsymbol{\varepsilon} \sim \mathcal{N}(\mathbf{0}, I_N),$$

with known $\boldsymbol{\mu}_t \in \mathbb{R}^N$ and *invertible* $\mathbf{L}_t \in \mathbb{R}^{N \times N}$. The goal is to predict $\varepsilon^{(N-1)}$.

By whitening trick, one can isolate $\boldsymbol{\epsilon}^{(N-1)}$. Let $\mathbf{e}_{N-1}^\top = [0, \ldots, 0, 1]$ be the row vector that selects the last coordinate. Left-multiplying by $\mathbf{e}_{N-1}^\top \mathbf{L}_t^{-1}$ gives

$$\mathbf{e}_{N-1}^\top \mathbf{L}_t^{-1} \mathbf{x}_t = \mathbf{e}_{N-1}^\top \mathbf{L}_t^{-1} \boldsymbol{\mu}_t \, x_1 + \mathbf{e}_{N-1}^\top \underbrace{\mathbf{L}_t^{-1} \mathbf{L}_t}_{I_N} \boldsymbol{\varepsilon}.$$

Define the *time-dependent scalars*

$$\mathbf{a}_t^\top := \mathbf{e}_{N-1}^\top \mathbf{L}_t^{-1} \in \mathbb{R}^N, \quad b_t := \mathbf{a}_t^\top \boldsymbol{\mu}_t \neq 0,$$

then

$$\boldsymbol{\epsilon}^{(N-1)} = \mathbf{a}_t^\top \mathbf{x}_t - b_t \, x_1. \tag{62}$$

A neural network $\varepsilon_\theta(\mathbf{x}_t, t)$ is trained to approximate $\varepsilon^{(N-1)}$ with the standard

$$\mathcal{L}_{\text{MDM}}(\theta) := \mathbb{E} \| \varepsilon_\theta(\mathbf{x}_t, t) - \boldsymbol{\epsilon}^{(N-1)} \|_2^2$$

Insert eq. 62 and multiply the interior by $b_t$:

$$\mathcal{L}_{\text{MDM}}(\theta) = \mathbb{E} \| \varepsilon_\theta(\mathbf{x}_t, t) - \mathbf{a}_t^\top \mathbf{x}_t + b_t \, x_1 \|_2^2$$

$$\propto \mathbb{E} \| g_\theta(\mathbf{x}_t, t) - x_1 \|_2^2$$

where

$$g_\theta(\mathbf{x}_t, t) := -\frac{\varepsilon_\theta(\mathbf{x}_t, t) - \mathbf{a}_t^\top \mathbf{x}_t}{b_t}$$

