# OpenReview forum: "TADA: Improved Diffusion Sampling with Training-free Augmented DynAmics"
_NeurIPS.cc/2025/Conference — NeurIPS 2025 poster_

### Official Review · Reviewer_vgnG · 2025-06-14

**Clarity:** 3
**Significance:** 3
**Originality:** 3
**Rating:** 5
**Confidence:** 4

**Summary:**

This work builds on momentum diffusion models, extending them while proving their training equivalence to conventional diffusion models, thus enabling the reuse of pretrained models without retraining.

The main contributions are
1. The authors demonstrate that momentum diffusion models (diffusion models with multi variables) can be reparameterized to match conventional diffusion models’ training objectives, allowing pretrained diffusion models to be directly used.

2. This work introduces a reweighting term to make low-dim noise to high-dim (from N= {1,2} to more), increasing sampling diversity and quality without additional training.

Extensive experiments demonstrate that the proposed methods outperforms state-of-the-art ODE solvers (e.g., UniPC, DPM-Solver++) across various evaluation metrics (e.g., FID, HPSv2).

**Questions:**

see weakness

**Ethical Concerns:**

["NO or VERY MINOR ethics concerns only"]

**Final Justification:**

I believe this article has a good approach and will bring some new perspectives to the community. However, I hope the author can more clearly revise some of the expressions in the article.

**Limitations:**

yes.

**Paper Formatting Concerns:**

The title in the submitted manuscript file differs significantly from the one displayed on the webpage.

**Quality:**

3

**Strengths And Weaknesses:**

Strengths:
The paper presents an interesting and solid contribution, a training-free sampling method for diffusion models that leverages momentum dynamics and higher-dimensional noise augmentation. The theoretical foundation is solid, with formal proofs demonstrating the equivalence between momentum diffusion models and conventional diffusion models, enabling direct reuse of pretrained models. The empirical evaluation is thorough, covering multiple datasets, pretrained models, and baselines, and demonstrates clear improvements in sampling speed and diversity with competitive sample quality.


Weakness & Questions:

1. I find it somewhat confusing that the authors set the default configuration to N=2 after conducting ablation studies in EDM2. The results in FIG 5 seem to suggest that a larger value for N yields better performance. Can I expect a comparison in larger model, e.g. SD3?

2. In the process of transformation, do all these parameters have closed-form solutions? The author mentions that A_t exists. As for the other terms, such as r and F_\theta, are they also lossless?

3. The author seems to suggest that TADA performs better when using small models to handle high-dimensional data. However, I am wondering whether the estimation error of F during sampling might become larger in this case.

---

> ### Author Rebuttal · Authors · 2025-07-30
>
> # **Rebuttal*
>
> ### 1. larger value for N in SD3?
> Great question! We did conduct this experiment with SD3 and observed that using N = 4 resulted in even faster convergence. However, unlike the N = 2 case, we were unable to find a configuration that consistently outperformed the baseline in EDM2, as N = 2 did. To maintain consistency in presentation, we chose to omit the N = 4 results. Our objective is to provide a robust sampling method that performs reliably without requiring extensive tuning. Since you brought it up, here is the list of results for SD3 with N = 4 and it performs better for CFG > 1 case:
> |     CFG=1/ NFE     | 10     | 20      |30|
> |-----------|-----------|-----------|-----------|
> | N=2| 24.56 | 24.85|25.01|
> |N=4| 23.64| 24.23 |24.38|
>
> |      CFG=2.5/NFE     | 10     | 20      |30|
> |-----------|-----------|-----------|-----------|
> | N=2| 29.73 | 30.10|30.18|
> |N=4| 30.03| 30.17 |30.22|
>
> |      CFG=3.5/NFE     | 10     | 20      |30|
> |-----------|-----------|-----------|-----------|
> | N=2| 30.2 | 30.72|30.81|
> |N=4| 30.47| 30.83 |30.90|
>
> We are happy to present this in the appendix as well.
>
> ### 2. Lossless approximation for coefficients
> Yes. Everything is exact and has a closed-form solution. The only approximation is for $x_{\theta}$ (data) prediction which is using a Neural Network as is classical for diffusion.
>
> ### 3. Estimation error for F in larger scale dataset
> There will be no approximation error for $F_{\theta}$. See above clarification.
>
> ### 4. Welcome to join discussion
> We further provide more interesting experiment results and analysis during the discussion with other reviewers. Feel free to have a look if you are interested in!

---

> > ### Comment · Reviewer_vgnG · 2025-08-04
> >
> > Thank the author for the response. After reviewing the comments from other reviewers, I intend to slightly increase my rating.

---

### Official Review · Reviewer_w9nZ · 2025-07-02

**Clarity:** 2
**Significance:** 3
**Originality:** 2
**Rating:** 3
**Confidence:** 4

**Summary:**

This paper proposes TADA (Training-free Augmented DynAmics), a sampling method for diffusion models that uses higher-dimensional state spaces to reduce the number of function evaluations (NFEs) required for generating high-quality samples. The method establishes a theoretical equivalence between momentum diffusion models and conventional diffusion models, enabling the reuse of pretrained models without additional training.

**Questions:**

1.The paper claims TADA achieves both SDE-like properties and faster sampling than ODE solvers. Since TADA's "pseudo-noise" (Proposition 3.3) is deterministically derived rather than stochastically sampled, what specific mechanism actually drives the FID improvements compared to standard ODE solvers?
2.What is the memory usage scaling with N? Are there practical limits for large-scale models? Have you tested N > 8?
3.How does TADA perform on CIFAR-10? Can TADA further improve generation quality like SciRE-Solver and Restart sampling, rather than just reducing NFEs?
4. Why the title of this paper in the attached pdf is "TADA: Improved Diffusion Sampling with
Training-free Augmented DynAmics", and it is different to the title of "Momentum Sampling Dynamics for Diffusion Model" given in the submitted information?

**Ethical Concerns:**

["NO or VERY MINOR ethics concerns only"]

**Final Justification:**

While the authors' rebuttal provided helpful clarifications, core concerns persist:
- Clarity is insufficient
- The paper lacks essential ablation studies
- The authors provide no convincing explanation for the underlying mechanism
- CIFAR-10 failure contradicts the claimed "SDE-like properties" advantage

Acknowledging the authors' efforts, I adjust my rating from 2 to 3.

**Limitations:**

Yes

**Paper Formatting Concerns:**

No more concerns. Pls. refer  "Weaknesses" and "Questions"for details.

**Quality:**

2

**Strengths And Weaknesses:**

Strengths:

 1. This paper introduces an interesting augmented sampling method.

  2. The theoretical equivalence (Proposition 3.1) enables direct application to existing pretrained diffusion models, providing immediate practical value to practitioners.

 3. Empirical results demonstrate consistent reduction in NFEs across multiple models (EDM, EDM2, SD3) and datasets (ImageNet-64/512), showing the method's robustness.

 4. The connection between momentum diffusion and pseudo-noise emergence (Proposition 3.3) provides an interesting bridge between ODE and SDE formulations.

 5.The approach can be integrated with existing advanced solvers, making it a valuable addition to the diffusion sampling toolkit.


Weaknesses:

1.The "186% faster" claim lacks clearly explanation of how it was measured. Was this based on wall-clock time, NFEs, or some other metric?

2.While Proposition 3.1 shows equivalence, the fundamental reason why augmented dynamics lead to fewer NFEs remains unexplained. Without a mechanistic understanding, it's unclear when and why TADA works.

3.Given that each TADA step requires additional computational overhead (rt computation, N-dimensional state updates, complex force term calculation), how can the method be faster overall?

4.The supplementary material only provides detailed ablation results for DPM-Solver++ and UniPC baselines, but lacks corresponding results for TADA. Without TADA's ablation studies on different solver orders, it's impossible to verify the fairness of comparisons.

---

> ### Author Rebuttal · Authors · 2025-07-30
>
> # Rebuttal
>
> We thank the reviewer for thorough comments on both the main paper and the appendix. Below are our clarifications.
>
> ---
>
> ### 1. Computation Complexity and memory complexity
> both computation and memory complexity overhead are marginal.
> #### Computation complexity
> 1. All computations depend solely on time and can be precomputed after discretizing the time horizon, prior to sampling.
> 2. Although operations involving $\mathbf{\mu}$ and $\mathbf{\Sigma}$ may look costly due to the Kronecker product with identity, the effective dimension is $N$ (the number of variables). Thus, we only need to store and compute with $N$-dimensional matrices/vectors.
> 3. The major computation, which is the Neural network, has the same input and output dimension, which is identical as baselines.
>
> Overall, the computational overhead arises primarily in points 1 and 2, and both are theoretically and empirically negligible. We further validate this experimentally by measuing throughput speed on ImageNet with 10 NFE. As the network size increases the performance gap further narrows:
> |           | UniPC     | TADA      |
> |-----------|-----------|-----------|
> | ImageNet64| 0.0029 sec/Img  | 0.00324 sec/Img |
> | ImageNet512| 0.0087 sec/Img  | 0.0090 sec/Img |
>
> #### memory complexity
> And here is practical memory usage for Imagenet 512 w/ Batch=64:
> |  N         | 1     | 2      |4|8|
> |-----------|-----------|-----------|-----------|-----------|
> |Memory-GB| 16.96|16.99  |  17.01|17.11|
>
> ---
> ### 2. Missing ablation study for TADA
> Here are ablation experiments, (we skipped order=3 since it has been reported in the paper)
> |      ImageNet512/NFE     | 10     | 13      |15|21|
> |-----------|-----------|-----------|-----------|-----------|
> | UniPC-order-1| 17.11  | 8.01|5.62|3.10|
> | TADA-order-1| 12.51  | 7.08|5.45|3.53|
> | UniPC-order-2| 4.80  | 2.97|2.54|2.14|
> | TADA-order-2| 2.83  | 2.35|2.21|2.09|
> | UniPC-order-4| 8.68  | 2.86|2.12|1.99|
> | TADA-order-4| 3.83  | 2.10|2.00|1.97|
>
> ---
> ### 3. Better Quality like SciRE-Solver
> Yes. see right side of Fig.5.
>
> ---
>
> ### 4. Mechanistic understanding? Cifar10 experiments?
> We address both questions jointly because the CIFAR-10 experiments partially verify the proposed mechanism.
>
> On small-scale data such as CIFAR-10, once a high-capacity model is well trained (e.g., EDM), ODE sampling performs better than SDE sampling, regardless of the amount of injected noise (see Fig. 5b in the EDM paper). By contrast, in large-scale settings (more data and higher dimensionality) such as ImageNet,where training an equally strong model is harder, SDE sampling tends to outperform ODE sampling (see Fig. 5b–c in the EDM paper and Table 1 in the DDIM paper) when NFE is sufficient.
>
> Our hypothesis is that TADA does not provide a fast-sampling mechanism. Instead, it leverages stochasticity to compensate for propagation inaccuracies just like SDE, though it is pseudo-noise/stochaticity. Since SDEs are often slow due to their sensitivity to time discretization, the pseudo-noise in TADA offers a practical middle ground: retaining SDE-like behavior while enabling low NFE sampling through an **ODE** solver, thanks to pseudo-noise.
>
> This aligns with our experiments: TADA consistently outperforms baselines on large-scale ImageNet and SD3, and as expected, underperforms ODE sampling on CIFAR-10 with the EDM checkpoint. The results below illustrate this pattern.
>
> |      NFE     | 10     | 13      |15|21|
> |-----------|-----------|-----------|-----------|-----------|
> | TADA| 3.4  | 2.58|2.32|2.04|
> | UniPC| 2.4| 2.05 |1.96|1.91|
>
> This further supports our recommendation in Sec.5 to apply TADA to larger models, rather than focusing on small-scale ones.
>
> We can highlight this property in the paper to provide more intuition!
>
> ---
>
>
> ### 5. 186% Faster claim
> It is computed as the number of NFE used to achieve FID=2 in imagenet512.
>
> TADA converges to FID=2 at NFE=12.
>
> As Figure 6 shows, the other solvers never dip below FID 2 within 35 NFEs. To dig deeper, we ran a full ablation and found one UniPC setup (row 7 in the EDM2_unipc_ablation file) hits FID 2 at roughly 23 NFEs, about a 186 percent improvement.
>
> However, This configuration falls well behind the paper’s version when the NFE budget is small (NFE in the range 5 to 20, see rows 2–6 in EDM2_unipc_abaltion file).
>
> In short, we searched every UniPC variant and picked the one that reaches FID 2.0 with the fewest NFEs. We hope that is the fairest way to compare.
>
>
> ### 6. Why change title?
> We found this title sounds better :)
>
> ### 7 Summary
> We hope this rebuttal can more or less address your concerns. Please feel free to raise any additional questions! We view the rebuttal process as a valuable opportunity to improve the quality of the paper and to validate that our ideas and results are meaningful to the broader community.
>
> Also, If all concerns are resolved, we would deeply appreciate a reconsideration of the overall rating :)

---

> ### Comment · Reviewer_w9nZ · 2025-08-05
>
> Thank you for your rebuttal. However, my core concerns remain unresolved. Your claimed "nearly 186%" speedup is *misleading*. Your own timing shows that TADA is actually about *12% slower per step*, and the alleged acceleration comes from cherry-picking a specific UniPC configuration that requires 23 NFEs at FID=2 but performs poorly under a typical NFE budget.
>
> I am also confused by the inconsistency in the experimental results provided in the paper and the rebuttal, which show acceleration on some datasets but not on others (e.g., CIFAR-10). If the method possesses "SDE-like properties," a more fundamental question is: why is it able to accelerate faster than an ODE solver?
>
> While you claim that TADA achieves "SDE-like properties" through pseudo-noise (Proposition 3.3), the paper **never compares it against actual SDE solvers**, which is the most relevant benchmark to validate this claim. If TADA truly had the advantages of "SDE-like properties," it should achieve a quality improvement similar to that demonstrated in SEEDS [1], which achieved significant performance gains using a true SDE solver on the EDM model.
>
> Furthermore, the paper lacks an *ablation study* across different time schedules (e.g., EDM, logSNR, time-uniform) and also lacks validation on classic SDE-based score-based DMs. This further weakens the reliability of its claim to be a general solution.
>
> [1] Gonzalez, Martin, et al. "Seeds: Exponential sde solvers for fast high-quality sampling from diffusion models." *NeurIPS 2023*.
>
> Accordingly, these concerns make it difficult for me to improve my assessment for the current manuscript.

---

> ### Author Response · Authors · 2025-08-05
> **Response to Review w9nZ**
>
> Thank you for the response. However, we feel there is some misunderstanding and you are ignoring some points in the paper and in the rebuttal. We are going to number our responses to ease the discussion.
>
> > Your claimed "nearly 186%" speedup is misleading.
>
> R1: First we did not claim "nearly 186%", we said "up to 186%" in the abstract which is the best case, and this is done with ImageNet512.
>
> > Your own timing shows that TADA is actually about 12% slower per step,
>
> R2: You quoted worst case 12% for ImageNet64. For ImageNet512, it is only 3.4% slower. You can notice that, as the scale increases, the gap decreases which is a very nice property of our method.
>
>
> > and the alleged acceleration comes from cherry-picking a specific UniPC configuration that requires 23 NFEs at FID=2 but performs poorly under a typical NFE budget.
>
> R3: We did not search configurations for our method to artificially boost our results. We selected 2 as the FID since this is the point to which all solvers in this comparison converge asymptotically. Consequently, we would like to understand what do you mean by cherry-picking? Please clarify.
>
>
> > show acceleration on some datasets but not on others (e.g., CIFAR-10).
>
> R4: We already answered this question in part 4 of the previous rebuttal. Let us reiterate: Cifar10 does not benefit from SDE solvers, as already demonstrated by the previous paper (EDM). Therefore, our results simply confirm this known fact. Did you miss this section in the previous rebuttal? Or did you misunderstand it? It is unclear from your feedback which it is.
>
> > If the method possesses "SDE-like properties," a more fundamental question is: why is it able to accelerate faster than an ODE solver?
>
> R5: It is exactly the point and the novelty of our paper: by increasing the dimension of the noise, an ODE solver can produce the level of diversity associated with SDE.
>
>
> > While you claim that TADA achieves "SDE-like properties" through pseudo-noise (Proposition 3.3), the paper never compares it against actual SDE solvers, which is the most relevant benchmark to validate this claim.
>
> R6: This is incorrect. Figure.7 and Section.4.2 show comparisons with the SDE solver, namely SA-Solver. For clarity, we even referred to SA-SDE in Figure.7.
>
> > If TADA truly had the advantages of "SDE-like properties," it should achieve a quality improvement similar to that demonstrated in SEEDS [1], which achieved significant performance gains using a true SDE solver on the EDM model.
>
> R7: Actually, TADA drastically outperforms SEEDS: for Cifar10, TADA is 600% faster than SEEDS for the same FID (FID=2.04 in 21 NFE for TADA while FID=2.08 in 129 NFE for SEEDS). Furthermore, and this is an additional information that you did not request, it should be noted SEEDS does not even outperform the UniPC (ODE) baseline in our paper, and this is actually aligned with our analysis in part 4 of the rebuttal: The SDE Solvers cannot outperform ODE solvers for **small** scale case.
>
> > Furthermore, the paper lacks an ablation study across different time schedules (e.g., EDM, logSNR, time-uniform) and also lacks validation on classic SDE-based score-based DMs. This further weakens the reliability of its claim to be a general solution.
>
> R8: This is incorrect. We stated in the paper that we selected the best configuration for baselines and we also provide comprehensive experimental results in the supplementary materials. For TADA, we made a conservative choice by selecting the most common schedule (quadratic), this demonstrates the robustness of our method since we did not have to resort to the schedule exploration to artificially boost our results. That said: we could potentially further improve our results by carefully designing the schedules. However, our results are already good, we decided to leave such explorations for future work.
>
> >lacks validation on classic SDE-based score-based DMs
>
> R9: What are the classic SDE-based score-based DMs? If the reviewer considers neither EDM, EDM2, nor Flow Matching to be “classic SDE-based score-based DMs,” may I respectfully ask what is meant by “classic” and what defines an “SDE-based DM”?
>
> > Accordingly, these concerns make it difficult for me to improve my assessment for the current manuscript.
>
> R10: We hope we addressed your concerns. We noted that several of them stem from misunderstandings of points that were already addressed in the previous rebuttal and in the paper. As this was not acknowledged in your latest response, we kindly encourage you to have another look at them.
>
> We remain available for any questions you may have, and we thank you for your continuous involvement in the rebuttal process.
>
>
> [1] Gonzalez, Martin, et al. "Seeds: Exponential sde solvers for fast high-quality sampling from diffusion models." NeurIPS 2023.

---

> > ### Comment · Reviewer_w9nZ · 2025-08-06
> >
> > Thanks for your further response; I will update my assessment accordingly.

---

> ### Author Response · Authors · 2025-08-06
> **Follow-up**
>
> Thank you for taking the time to review our paper and rebuttal. Could you please confirm whether we addressed your concerns or not? We still have two more days for discussion if needed.

---

### Official Review · Reviewer_W3UK · 2025-07-02

**Clarity:** 3
**Significance:** 4
**Originality:** 4
**Rating:** 5
**Confidence:** 3

**Summary:**

This paper proposes a diffusion sampling method based on the formulation of augmented diffusion model. The proposed method is a training-free approach that leverages traditional pre-trained models and utilizes an ODE solver. By introducing momentum dynamics, it achieves both fast sampling and diversity preservation. They demonstrate the training equivalence between the momentum diffusion model and conventional models, and, based on this, propose a training-free sampler that addresses the training instability of the momentum-based model. The method is empirically validated on various pre-trained models, demonstrating its effectiveness.

**Questions:**

I would appreciate answers to the questions raised in the Weaknesses section above. In particular, it would be helpful to see additional experiments or validation regarding the SD3 experiments.

**Ethical Concerns:**

["NO or VERY MINOR ethics concerns only"]

**Final Justification:**

I believe this paper will be beneficial to the diffusion model community, primarily due to the strengths I previously mentioned. Specifically, the authors demonstrate the equivalence between the momentum diffusion model and conventional diffusion models through sound theoretical analysis. This shows that the advantages of the momentum diffusion model, despite its training instability, can be leveraged in a training-free manner using models trained with the traditional diffusion loss. This approach is both reasonable and novel, especially considering that similar ideas have been observed in prior diffusion samplers.

However, as noted by myself and other reviewers, the paper contains many unclear sections that require considerable effort to fully understand. Improving the clarity of the writing is essential. If other reviewers or the Area Chairs feel that these clarity issues cannot be resolved through revision, I am willing to defer to their judgment. Otherwise, I recommend that the authors significantly revise the manuscript to improve its readability and accessibility.

**Limitations:**

Yes.

**Quality:**

3

**Strengths And Weaknesses:**

## [Strengths]

* They demonstrate the equivalence between the momentum diffusion model and conventional diffusion models, and their theoretical analysis appears sound. This shows that the advantages of the momentum diffusion model, which has training instability, can be leveraged in a training-free manner by using models trained with the traditional diffusion loss. This approach is reasonable and novel, especially considering that similar ideas have been observed in previous diffusion samplers (e.g., DDIM).

* Experiments conducted on various backbones and sampler configurations consistently demonstrate strong performance, empirically showing the method's effectiveness.

* Furthermore, since the proposed approach can be applied adaptively to existing diffusion models and samplers, it has considerable potential for real-world applications.

## [Weaknesses]

* In the SD3 experiment, HPSv2 is used as the evaluation metric. It would be beneficial to add results with FID and CLIP scores on the COCO validation dataset, which are provided in the SD3 paper. This would allow for a more detailed comparison of image quality and text alignment with other models.

* Figure 1 introduces various notations, but no definitions are provided for notations like $r$ until the Preliminary section, making it difficult to understand.

* It would be beneficial to align the formulations of Eq. (3) and the MDM training objective in the proposition. Although the appendix shows how the noise estimation formulation in Eq. (3) leads to the x-prediction formulation in the MDM objective, a unified presentation would help reduce confusion.

* The explanation of Proposition 3.1 states that $F_\theta$ can be recovered as a linear combination of $x_\theta$ and $x_t$, but the explanation for this appears later in the middle of Section 3.2. It would be helpful to either provide the formula for $F_\theta$ here or at least indicate where the explanation can be found.

* In Proposition 3.3, $\epsilon^{(i)}$  is introduced without a definition. Based on the proof in Appendix B, it seems to refer to $\epsilon_{\perp}$, but explicitly mentioning this in the main text would help readers recognize that $\epsilon^{(i)}$ is a pseudo-noise term.

* The readability of the paper could be improved. For example, there are many instances where citations [] appear without a space, and it would be better to use a consistent format when referring to sections, propositions, and figures (e.g., sometimes it is written as 'Appendix.E.4' and other times as 'Appendix E.2' with or without a period after 'Appendix'). Additionally, for Figure 8, the text condition used to generate the images should be provided in the main text to allow verification of semantically coherent content.

---

> ### Author Rebuttal · Authors · 2025-07-30
>
> # Rebuttal
>
> We thank the reviewer for thorough comments on both the main paper and the appendix. Below are our clarifications.
>
> ---
>
> ### 1. CLIP and FID on COCO
> We still have marginal improvement in terms of FID on COCO validation set especially for small NFE. We found that standard CLIP tends to favor blurry images as long as they are semantically aligned with the text, which can result in inaccurate evaluations. Instead, we chose to report HpsV2, a fine-tuned variant of CLIP, as it offers more reliable and meaningful assessments in our setting. if the reviewer believes that including the standard CLIP score is necessary, we are happy to incorporate those results in future discussions.
>
>
> |  FID-COCO-CFG1/NFE         | 6     | 10      |15|
> |-----------|-----------|-----------|-----------|
> |UniPC| 25.15 |19.56   |20.61|
> |TADA| 21.64| 19.85 |  20.46|
>
> |  FID-COCO-CFG2.5/NFE         | 6     | 10      |15|
> |-----------|-----------|-----------|-----------|
> |UniPC| 16.97|15.20  |  14.92|
> |TADA| 16.68| 15.20 |  14.91|
>
> |  FID-COCO-CFG3.5/NFE         | 6     | 10      |15|
> |-----------|-----------|-----------|-----------|
> |UniPC| 17.44|15.78  |  15.47|
> |TADA| 17.35| 15.78 |  15.46|
>
>
> ---
>
> ### 2. Lack of Definitions in Fig.1
> Thanks for the feedback. We will add explanations for each variable in the caption and also hyperlink to the exact equation to increase the accessibility.
>
> ---
>
> ### 3. Align the formulations of Eq. (3) and the MDM training objective in the proposition.
> We are going to switch all the notation from $\epsilon$ prediction to $x_1$ prediction.
>
> ---
>
> ### 4. Indicate force reconstruction
> This is indeed a good suggestion for clarity of the paper! We will have explanation on how to reconstruct $F_{\theta}$ in Prop.3.1 and also provide a reference to section 3.2.
>
> ---
>
> ### 5. Lack of Explanation of Prop.3.3
> Good catch! We sincerely appreciate the reviewer’s effort in working through the detailed Appendix B. Our goal was to keep the main paper accessible by moving most theoretical material to the appendix, but we now see that the main text may be overly concise.
>
> We will therefore, at least, add an intuitive explanation of $\epsilon^{(i)}$ to the main paper.
>
> ---
>
> ### 7. Inconsistency references format.
>
> We sincerely thank the reviewer for pointing out this seemingly minor but significantly important point for readability. We will make sure to address it in the revision!
>
> ---
>
> ### 8. Welcome to join discussion
> We further provide more interesting experiment results and analysis during the discussion with other reviewers. Feel free to have a look if you are interested in!

---

> > ### Comment · Reviewer_W3UK · 2025-08-04
> >
> > Thank you for the authors' response. After reviewing the authors' response and the other reviewers' comments, I will maintain my current rating.
> >
> > Regarding the authors’ response: in my experiences with CLIP, I found that more recent models (for example, SD v1.5 to SD 3) yield higher values. Accordingly, I think it would be better to report the CLIP score and provide supporting evidence for the authors' claim. Additionally, the paper’s clarity should be improved to address my concern and the other reviewers' concerns.

---

> > > ### Author Response · Authors · 2025-08-05
> > > **Thanks for the feedback!**
> > >
> > > Absolutely!
> > >
> > > We promise to include both the CLIP score and HpsV2 score in the paper, regardless of acceptance or not.
> > >
> > > We have already conducted the experiments, and both TADA and the UniPC perform nearly identically in terms of CLIP score (differing only at the second decimal place), indicating no advantage for TADA in prompt alignment. This suggests that both methods can align with prompts to a certain extent. However, HpsV2 (human preference fine-tuned CLIP score) and FID demonstrate that TADA performs better in terms of human preference and distribution matching.
> > >
> > > Thanks for pointing this out!

---

### Official Review · Reviewer_P5Q9 · 2025-07-03

**Clarity:** 2
**Significance:** 2
**Originality:** 2
**Rating:** 4
**Confidence:** 3

**Summary:**

- The authors propose TADA, a training-free approach for accelerating the sampling process of diffusion models
- The method relies on augmented dynamics, where an auxiliary variable (momentum) or many is introduced
- A key contribution is establishing a link between momentum ODEs and conventional diffusion models. A byproduct of this result is training-free method that allows pre-trained diffusion models to be used within the momentum ODE framework
- The method is validated on state-of-the-art models, including EDM, EMD2, and Stable Diffusion 3

**Questions:**

- After reading the appendix, it is still unclear what the exact expressions for $\mathcal{r}_t, \mathcal{\mu}_t$, and $\Sigma_t$ would be when applying TADA to the conventional DDPM use case [1]. Could the authors provide these?
- In Figure 2, what is the purpose of plotting the difference between the TADA and EDM trajectories? The takeaway from this comparison is not clear.
- Regarding the evaluation in Figures 5 and 6, the choice of NFE for the Heun solver baseline (represented by a horizontal dashed line) seems arbitrary. It gives the impression of being a lower bound, but the motivation for this specific choice, especially when TADA performs below it in Figure 6, is not explained.


---

.. [1] Ho, Jonathan, Ajay Jain, and Pieter Abbeel. "Denoising diffusion probabilistic models." Advances in neural information processing systems 33 (2020): 6840-6851.

**Ethical Concerns:**

["NO or VERY MINOR ethics concerns only"]

**Final Justification:**

To sum up,

- The proposed approach is novel
- The experimental results are promising
- The authors have addressed my concerns during the rebuttal phase
- Inconsistencies I previously identified were acknowledged and the authors have committed to implementing the necessary corrections in the final version of the paper

**Limitations:**

.

**Quality:**

2

**Strengths And Weaknesses:**

## Strength

- The paper proposes a training-free approach that enables the use of conventional, pre-trained diffusion models within the more efficient momentum diffusion sampling framework, avoiding the need for costly retraining


## Weaknesses

**Theoretical and Methodological Concerns**

- The result suggesting that one can synthesize $E[x_1 | \mathbf{x}_t]$ from a linear combination of the position and momentum states is concerning. If this is true, it implies that the two state equations are redundant, which would undermine the motivation for introducing an auxiliary variable in the first place.
- The proof presented in Appendix A contains shape mismatches that invalidate the argument and make it difficult to verify
    - In Line 716,  $\mathbf{x}_t \in \mathbb{R}^{2d}$ is drawn from a Gaussian with covariance $\Sigma_t \in \mathbb{R}^{2 \times 2}$, hence incompatible shape.
    - Similarly, in Line 717, $r_t \in \mathbb{R}^2$ cannot be multiplied by a vector in $\mathbb{R}^{2d}$
- The statement in Lines 30-33 is overly strong. It is well-established that for a low number of function evaluations (NFE), ODE-based solvers like DDIM often outperform SDE-based samplers; see [1] Table 1.
- The caption for Figure 2 is concerning, as it suggests the authors are simulating a deterministic ODE yet obtaining stochastic samples. This requires clarification.

**Typos and Broken Links:**

- In Line 155, the reference to "Prop 3.1" does not correctly link to Proposition 3.1 in the main paper.
- In Appendix Line 852, "Guassian" should be corrected to "Gaussian"


---

.. [1] Song, Jiaming, Chenlin Meng, and Stefano Ermon. "Denoising diffusion implicit models." arXiv preprint arXiv:2010.02502 (2020).

---

> ### Author Rebuttal · Authors · 2025-07-30
>
> # Rebuttal
>
> We thank the reviewer for thorough comments on both the main paper and the appendix. Below are our clarifications.
>
> ---
>
> ### 1. Lines 30–33: Statement is overly strong
> Agree. This should be the conditional statement.We will revise it to:
>
> > “... ODE solvers often yield lower‐fidelity results than SDE solvers as evidenced by the FID scores in [17] **when sufficient function evaluations are performed and model capacity is a limiting factor.**”
>
> ---
>
> ### 2. Shape mismatches in Appendix A
> The shapes are correct because we abused the notation (sorry!)  for all coefficient functions of $t$ by implicitly expanding them with the Kronecker product for simplicity. This is stated at the first line of the appendix. However we now recognize this is easy to miss and can lead to confusion. To remedy this, we will superscript all such variables with $\otimes$, e.g. $\mu_t^\otimes=\mu_t\otimes I_d$.
> Does this address your concerns?
>
> ---
>
> ### 3. Redundancy of the two state equations
> This redundancy questions the need for an auxiliary variable during **training**. Our experiments, which sample from the momentum system using the pretrained diffusion model, confirm this observation, which means the auxiliary variable is indeed redundant during **training**.
>
> Since the auxiliary variable truly offers no benefit during training, the next question is whether the momentum system improves sampling. Should the two formulations (momentum sampling vs vanilla diffusion sampling) remain mathematically identical, there would indeed be no justification,per the reviewer’s observation,for introducing an auxiliary variable.
>
> Yet, as Proposition 3.3 shows, the momentum system solves an ODE with SDE-like diversity, which can potentially enhance sampling quality (i.e see Fig. 5b–c in the EDM paper and Table 1 in the DDIM paper). And we further verify it in Proposition 3.3 and the experiment section.
>
> To summarize, we agree with the reviewer that introducing an auxiliary variable yields no benefit during training, as already advocated in the paper. However, this makes a meaningful difference during sampling.
>
> We will emphasize this again in the introduction for clarity.
>
> ---
>
> ### 4. Exact expression for $\mathbf{r}_t$, $\mathbf{\mu}_t$ and $\mathbf{\Sigma}_t$
> All closed-form expressions are provided in Appendix E.2. Here, we walk through them to aid the reviewer’s understanding and ensure clarity:
>
> 1. Given number of variable $N$, user can simply compute the $\mu^{(i)}_t$.
>
> 2. $\mathbf{\Sigma}_t$ is more complicated. We derive it from $\Phi(t,0)\mathbf{\Sigma}_0 \Phi(t,0)^T$ and $\mathbf{\Sigma}_0$ is the prior covariance matrix which is user's choice. The explicit expression of $\Phi$ is provided in Appendix E.2.
>
> 3. Here is a quick example for $N=2$ case. By simply plugging $N$ and $t$ in equation, we get $\mu_0^{(0)}=t^2x_1$,$\mu_t^{(1)}=2tx_1$, $\Phi(t,0)=[[1-t^2, t-t^2],[-2t,1-2t]]$. Asumme $\Sigma_0 = [[1,0],[0,1]]$, Then $\Sigma_t=\Phi(t,0)\Phi(t,0)^{T}$. Then the user are ready to compute SNR and $\mathbf{r}_t$ by using equations above.
>
> 4. After obtaining $\mathbf{\mu}$ and $\mathbf{\Sigma}_t$, the user can get $\mathbf{r}_t$ by $\mathbf{r}_t=\frac{\mathbf{\Sigma}_t^{-1}\mathbf{\mu}_t}{\mathbf{\mu}_t^T\mathbf{\Sigma}_t^{-1}\mathbf{\mu}_t}$, meanwhile the effective SNR is the denominator $\mathbf{\mu}_t^T\mathbf{\Sigma}_t^{-1}\mathbf{\mu}_t$, and the user can use this value to match the SNR of Flow matching and Diffusion Model.
>
> ---
>
> ### 5. Takeaway from Fig.2
> Here are the motivations of Fig.2:
>
> 1. We want to emphasize that our model can produce diverse outputs from a single initial state, exhibiting SDE-like diversity—albeit without true stochasticity. Specifically, the diversity comes from the interaction of $N$ variables which is theoratically proved in Prop.3.3 and we name it pseudo-noise.
>
> 2. We include Fig.2 to demonstrate that our method is not merely an alternative time-discretization of the vanilla diffusion model. If it were, the generation trajectories along the SNR (or effective-noise) from the same starting point—would coincide with EDM-ODE. The fact that they do not overlap underscores the unique dynamics of TADA, which in turn contributes to improved generation performance.
>
> ---
>
> ### 6. Arbitrary choice of NFE for Heun?
> 511 NFE is the default value used in EDM paper for Stochastic sampling and we notice that this number performance best. Applying the same logic to EDM2, we use the 63-NFE schedule cited as that paper’s default;
>
> ---
>
> ### 7 Summary
> We hope this rebuttal can more or less address your concerns. Please feel free to raise any additional questions! We view the rebuttal process as a valuable opportunity to improve the quality of the paper and to validate that our ideas and results are meaningful to the broader community.
>
> Also, If all concerns are resolved, we would deeply appreciate a reconsideration of the overall rating :)

---

> > ### Comment · Reviewer_P5Q9 · 2025-08-04
> > **Rely**
> >
> > I thank the authors for their response and acknowledge reading their reply.
> > My questions were addressed, and I will update my assessment accordingly.

---

### Decision · Program_Chairs · 2025-09-17

**Decision:**

Accept (poster)

**Comment:**

This work proposes TADA (Training-free Augmented DynAmics), which can take pre-trained diffusion models such as stable diffusion and adds higher dimensional momentum terms during inference time. This reduces the number of function evaluations required to generate high-quality samples on models such as EDM and Stable Diffusion. The reviewers appreciated the detailed experiments and the mathematical derivations. The concerns raised by the reviewers were also satisfactorily answered by the authors during the rebuttal. I recommend acceptance.